# Training-Free Adaptation of Diffusion Models via Doob's $h$-Transform

Qijie Zhu [* 1]   Zeqi Ye [* 2]   Han Liu [1 3]   Zhaoran Wang [2]   Minshuo Chen [2]

## Abstract

Adaptation methods have been a workhorse for unlocking the transformative power of pre-trained diffusion models in diverse applications. Existing approaches often abstract adaptation objectives as a reward function and steer diffusion models to generate high-reward samples. However, these approaches can incur high computational overhead due to additional training, or rely on stringent assumptions on the reward such as differentiability. Moreover, despite their empirical success, theoretical justification and guarantees are seldom established. In this paper, we propose `DOIT` (**D**oob-**O**riented **I**nference-time **T**ransformation), a training-free adaptation method for generic, non-differentiable rewards. We develop two variants of this method: `DOIT-Proto`, a full-rollout simulation-based algorithm and `DOIT-Practical`, an efficient surrogate algorithm. The key idea is a measure transport formulation that seeks to transport the pre-trained generative distribution to a high-reward target distribution. We leverage Doob's $h$-transform to realize this transport, which induces a dynamic correction to the diffusion sampling process and admits simulation-based computation without modifying the pre-trained model. Theoretically, we establish a high-probability convergence guarantee to the target high-reward distribution for `DOIT-Proto` by characterizing the approximation error in the dynamic Doob correction. Empirically, `DOIT-Practical` consistently outperforms state-of-the-art baselines on D4RL offline RL benchmarks while preserving sampling efficiency.

---

[*]Equal contribution [1]Department of Statistics and Data Science, Northwestern University, Evanston, IL, USA [2]Department of Industrial Engineering and Management Sciences, Northwestern University, Evanston, IL, USA [3]Department of Computer Science, Northwestern University, Evanston, IL, USA. Correspondence to: Minshuo Chen <minshuo.chen@northwestern.edu>.

*Proceedings of the 43$^{rd}$ International Conference on Machine Learning*, Seoul, South Korea. PMLR 306, 2026. Copyright 2026 by the author(s).

## 1. Introduction

Diffusion models have recently become a leading class of generative models, achieving state-of-the-art performance across a wide range of applications, including image generation (Song & Ermon, 2019; Song et al., 2020a;b; Ho et al., 2020; Kong et al., 2021; Jeong et al., 2021; Mittal et al., 2021; Huang et al., 2022; Ulhaq & Akhtar, 2022; Avrahami et al., 2022), molecular design (Weiss et al., 2023; Guo et al., 2024), and robotics (Chi et al., 2025; Reuss et al., 2023; Scheikl et al., 2024; Hou et al., 2024; Dasari et al., 2025). Notably, pre-trained diffusion models already exhibit strong capabilities. They can generate high-fidelity, photorealistic images (Song & Ermon, 2019; Ho et al., 2020; Song et al., 2020a;b; Nichol et al., 2022; Yang et al., 2024b), and in robotics, imitation-trained diffusion policies can produce reasonable action sequences for manipulation (Chi et al., 2025; Ze et al., 2024; Scheikl et al., 2024).

Despite the tremendous success, there is a pressing need to adapt pre-trained models to specific downstream tasks. For example, in robotics, diffusion models can serve as powerful and flexible policy classes. When trained on offline expert demonstrations, they can closely mimic expert behavior by generating action sequences consistent with the demonstrations (Chi et al., 2025), yet still underperform on the downstream task objectives that require action generation beyond imitation (Ren et al., 2025; Ada et al., 2024). These downstream task objectives can often be summarized as an abstract scalar-valued reward function $r(x)$ on the generated data, where a high reward value is desired. For instance, in Reinforcement Learning (RL) and robotics, the reward $r$ is often tied to task completion (e.g., reaching a target position). Given the reward function, adapting a pre-trained diffusion model amounts to steering its sample generation toward high rewards.

Many methods have been developed to adapt a pre-trained diffusion model toward high rewards. An ideal adaptation algorithm should be computationally lightweight, not data-hungry in terms of additional samples or interactions, and admit meaningful performance guarantees. One line of work focuses on training-based methods, including RL-based fine-tuning (Clark et al., 2024; Black et al., 2024; Fan et al., 2023; Prabhudesai et al., 2023; Uehara et al., 2024a;b; Ren et al., 2025; Hu et al., 2025; Zhao et al., 2025a; Liu et al., 2026),

guidance-based methods (Dhariwal & Nichol, 2021) and preference-based fine-tuning such as Direct Preference Optimization (DPO) (Wallace et al., 2024; Yang et al., 2024a; Lee et al., 2025). These methods substantially improve reward-aligned performance, at the cost of additional network training and hyperparameter tuning.

A complementary line of work develops training-free methods, which require no additional training, and can still achieve competitive performance. Some approaches assume access to the gradient of the reward (Chung et al., 2023; Bansal et al., 2023; Yu et al., 2023; Ye et al., 2024; Nguyen et al., 2025). This assumption, however, often fails in practice—for example, the reward function in molecular design often comes from black-box tools or discrete descriptors, which are non-differentiable (Trott & Olson, 2010; Abramson et al., 2024). To address this, several methods rely only on querying reward values, which include Sequential Monte Carlo (SMC)-based methods (Trippe et al., 2022; Wu et al., 2023; Dou & Song, 2024; Cardoso et al., 2024; Phillips et al., 2024; Kim et al., 2025) and search-based methods (Li et al., 2024; Ma et al., 2025; Li et al., 2025; Zhang et al., 2025a; Jain et al., 2026; Zhang et al., 2025b). While avoiding additional network training, these methods increase inference-time cost and are prone to sample collapse (Browne et al., 2012; Uehara et al., 2025). Such limitations motivate the following key questions:

*Can we design a training-free inference-time adaptation method for non-differentiable rewards?*
*If so, can such a method be theoretically grounded while remaining computationally efficient?*

We provide positive answers to these two questions and present an inference-time adaptation method `DOIT` (**D**oob-**O**riented **I**nference-time **T**ransformation). Specifically, suppose that a pre-trained diffusion model yields a generative distribution $P_\theta$, where $\theta$ denotes the pre-trained parameters in the score network. `DOIT` formulates the adaptation task as sampling from a conditional distribution $P_\theta(\cdot|\mathcal{E})$, where the event $\mathcal{E}$ encapsulates desired conditions, e.g., the reward of generated samples is beyond a threshold $r_0$. It leverages Doob's $h$-transform (Rogers & Williams, 2000; Särkkä & Solin, 2019) for the measure transport from $P_\theta$ to $P_\theta(\cdot|\mathcal{E})$. Importantly, Doob's $h$-transform introduces an additive correction term to the dynamic process of sample generation in diffusion models, allowing efficient implementation by keeping the pre-trained parameter $\theta$ frozen. Based on this method, we develop two instantiations: `DOIT-Proto`, a full-rollout simulation-based algorithm, and `DOIT-Practical`, an efficient surrogate implementation. We summarize our methodological, theoretical, and empirical contributions as follows.

● Methodologically, we propose a simulation-based ap-

proximation to the additive correction term in Doob's $h$-transform, leading to `DOIT-Proto`, a full-rollout algorithm. Building on `DOIT-Proto`, we further propose `DOIT-Practical` (Algorithm 2), an efficient surrogate version that is completely training-free, applies to non-differentiable reward functions, and maintains sampling efficiency comparable to pre-trained models.

● Theoretically, we provide a high-probability convergence guarantee for `DOIT-Proto`. Our analysis characterizes the error stemming from the approximation of the additive correction term in Lemma 5.2, and then we propagate the approximation error to establish an end-to-end total-variation bound between the output distribution of `DOIT-Proto` and the reward-induced target distribution in Theorem 5.4.

● Empirically, we demonstrate that `DOIT-Practical` effectively steers the distribution of generated samples toward high-reward regions in Section 6. On offline RL benchmarks, `DOIT-Practical` consistently outperforms state-of-the-art baselines while maintaining competitive sampling efficiency.

## 2. Related Work

**Training-Based Adaptation Methods** Reward adaptation is often achieved by updating diffusion model parameters with additional training. One approach casts the denoising process as a Markov decision process and applies RL-based methods to fine-tune pre-trained models for optimizing the reward (Clark et al., 2024; Black et al., 2024; Fan et al., 2023; Prabhudesai et al., 2023; Uehara et al., 2024a;b; Ren et al., 2025; Hu et al., 2025; Zhao et al., 2025a; Liu et al., 2026). Guidance-based methods learn a guidance term dependent on the reward function and distill the guidance into pre-trained model parameters (Ho & Salimans, 2022; Zhang et al., 2023; Yuan et al., 2023; Zhao et al., 2025b). More recently, preference-based methods such as DPO adapt diffusion models using pairwise comparisons (Wallace et al., 2024; Yang et al., 2024a; Lee et al., 2025). Despite strong empirical gains, these methods typically incur nontrivial training cost and may require additional data, interaction, or careful hyperparameter tuning.

**Training-Free Adaptation Methods** Training-free adaptation methods steer a pre-trained diffusion model at inference time. Several methods utilize the gradient of the reward function to guide the denoising sample generation process toward higher reward (Chung & Ye, 2022; Bansal et al., 2023; He et al., 2024; Yu et al., 2023; Ye et al., 2024; Nguyen et al., 2025). For non-differentiable reward functions, a growing set of approaches relies on inference-time scaling. A representative class is SMC-based methods (Trippe et al., 2022; Wu et al., 2023; Dou & Song, 2024; Kim et al., 2025; Singhal et al., 2025). They maintain a set

of particles, reweight them using reward information, and resample to concentrate on high-reward regions. Another class is search-based adaptation methods (Li et al., 2024; Ma et al., 2025; Li et al., 2025). Many of them perform local search by generating multiple denoising trajectories at each step and selecting the best one. More recent variants combine tree search with local search methods (Zhang et al., 2025a; Jain et al., 2026; Zhang et al., 2025b).

**Doob's $h$-transform Based Adaptation Methods**  There are several recent works leveraging Doob's $h$-transform to steer diffusion models toward a reward-induced target distribution. These works modify the sampling process by a correction term, which is learned by additionally training a neural network (Denker et al., 2024; 2025; Chang et al., 2026). A training-free method is developed in Nguyen et al. (2025) for text-to-image editing. Yet, it requires a differentiable reward function. Our method is training-free and applies to generic, non-differentiable rewards. Moreover, we establish a theoretical convergence guarantee of our method, which is highly limited with only very recent advances (Guo et al., 2026b; Chang et al., 2026).

**Notation**  For a vector $x$, let $\|x\|_2$ denote its Euclidean norm. Let $\|x\|_1$ and $\|x\|_\infty$ denote its $\ell_1$-norm and $\ell_\infty$-norm, respectively. For a matrix $A$, let $\|A\|_2$ denote its spectral norm. We use $\mathcal{O}(\cdot)$ to hide multiplicative constants in upper bounds. Unless otherwise stated, $\nabla$ denotes the gradient with respect to $x$; when there is no ambiguity, we drop $x$ from the notation.

# 3. Diffusion Model and Doob's $h$-transform

We briefly review the continuous-time formulation of diffusion models and their induced discrete-time sampling SDE (Section 3.1), and then introduce Doob's $h$-transform (Section 3.2) that will be used throughout the paper.

## 3.1. Diffusion Model Basics

A diffusion model aims to learn and sample from an unknown data distribution $P_{\text{data}}$ by estimating the score function (Song & Ermon, 2019; Ho et al., 2020; Song et al., 2020a). It consists of coupled forward and backward processes. The forward process is governed by the SDE:

$$\mathrm{d}Y_t = -\frac{1}{2}Y_t\mathrm{d}t + \mathrm{d}W_t \quad t \in [0,T], \quad Y_0 \sim P_{\text{data}}, \quad (1)$$

where $T$ is a terminal time, and $W_t$ is a Wiener process. We denote $P_t$ as the marginal distribution of $Y_t$ with density $p_t$. The backward process reverses the evolution in the forward process—referred to as denoising for new sample generation. Formally, the backward process is written as the reverse-time SDE:

$$\mathrm{d}X_t = \left[-\frac{1}{2}X_t - \nabla \log p_t(X_t)\right]\mathrm{d}t + \mathrm{d}\overline{W}_t, \quad (2)$$

where $t \in [0,T]$, $X_T \sim P_T$, and the SDE evolves backward in time from $T$ to $0$. Here, $\overline{W}_t$ denotes an independent Wiener process, and $\nabla \log p_t$ is the score function. As a result, we have $X_0 \sim P_{\text{data}}$.

In practice, the reverse-time SDE (2) is intractable due to the unknown terminal distribution $P_T$ and the unknown score function $\nabla_x \log p_t(x)$. Following standard practice, we replace $P_T$ by $\mathcal{N}(0,I)$ and $\nabla \log p_t$ by a trained score network $s_\theta(x,t)$. This leads to the following SDE:

$$\mathrm{d}\widetilde{X}_t = \left[-\frac{1}{2}\widetilde{X}_t - s_\theta(\widetilde{X}_t,t)\right]\mathrm{d}t + \mathrm{d}\overline{W}_t. \quad (3)$$

To simulate process (3), we discretize the time interval $[0,T]$ using a grid $0 = t_0 < t_1 < \cdots < t_{L-1} < t_L = T$, where $L$ is the number of discretization steps. Over each interval $t \in [t_{l-1}, t_l]$, we consider the piecewise SDE:

$$\mathrm{d}\bar{X}_t = \left[-\frac{1}{2}\bar{X}_t - s_\theta(\bar{X}_{t_l}, t_l)\right]\mathrm{d}t + \mathrm{d}\overline{W}_t, \quad (4)$$

which admits an analytical solution and allows for efficient sampling. Specifically, the transition distribution from $\bar{X}_{t_l}$ to $\bar{X}_{t_{l-1}}$ in (4) is

$$\bar{X}_{t_{l-1}}|\bar{X}_{t_l} = x_{t_l} \sim \mathcal{N}\left(\mu_{t_l}(x_{t_l}, s_\theta), \sigma_{t_l}^2 I\right), \quad (5)$$

where $\mu_{t_l}$ is a linear function of $s_\theta$ and $\sigma_{t_l}^2$ is a noise schedule. To enable fast sampling, existing literature modifies $\mu_{t_l}$ and $\sigma_{t_l}^2$, such as DDIM (Song et al., 2020a) and Euler ancestral sampling (Karras et al., 2022); see Appendix A for explicit forms. We denote $\bar{X}_0 \sim P_\theta$ as the generated distribution of (4). Furthermore, we denote $\bar{X}_t \sim P_{\theta,t}$ and let $p_{\theta,t}$ be its marginal density.

## 3.2. Doob's $h$-Transform

Recall that we view adapting a pre-trained diffusion model as steering sample generation toward high rewards. However, simply performing reward maximization can lead to reward hacking, where generated samples suffer from severe drops in fidelity and diversity (Skalse et al., 2022). To mitigate this, it is critical to model the conditional data distribution rather than pursuing pure reward maximization. Formally, we aim to sample from the conditional generated distribution $P_\theta(\cdot|\mathcal{E}_{\bar{X}_0})$, where $\mathcal{E}_{\bar{X}_0}$ describes the conditions on samples from the terminal distribution, with $\mathbb{P}(\mathcal{E}_{\bar{X}_0}) > 0$. A commonly used choice of $\mathcal{E}_{\bar{X}_0}$ is to select high-reward samples as

$$\mathcal{E}_{\bar{X}_0} = \{\bar{X}_0 : r(\bar{X}_0) \geq r_0\},$$

where $r_0$ is a threshold. Note that $P_\theta(\cdot|\mathcal{E}_{\bar{X}_0})$ is an approximation to the ideal conditional data distribution $P_{\text{data}}(\cdot|\mathcal{E}_{X_0})$ with $X_0$ the terminal state of (2), where $\mathcal{E}_{X_0}$ is defined analogously by replacing $\bar{X}_0$ with $X_0$.

**Transport $P_\theta$ to $P_\theta(\cdot|\mathcal{E}_{\bar{X}_0})$.**  Doob's $h$-transform (Rogers & Williams, 2000; Särkkä & Solin, 2019) provides a principled probabilistic framework to modify the generated distribution $P_\theta$ towards the desired conditional distribution

$P_\theta(\cdot|\mathcal{E}_{\bar{X}_0})$. The key is to define a Doob's $h$-function that dynamically modifies (4).

**Definition 3.1** (Doob's $h$-function)**.** The Doob's $h$-function for $0 \le t \le T$ is defined as

$$h(x_t, t) = \mathbb{P}(\mathcal{E}_{\bar{X}_0}|\bar{X}_t = x_t),$$

where $\mathbb{P}$ is taken with respect to the randomness in (4) and the randomness in $\mathcal{E}_{\bar{X}_0}$.

By Bayes' rule, the Doob's $h$-function induces a tilted density at time $t$:

$$p^h_{\theta,t}(x_t) = p_{\theta,t}(x_t|\mathcal{E}_{\bar{X}_0}) = h(x_t, t)p_{\theta,t}(x_t)/\mathbb{P}(\mathcal{E}_{\bar{X}_0}). \quad (6)$$

In particular, $p^h_{\theta,0}(x) = p_{\theta,0}(x|\mathcal{E}_{\bar{X}_0})$, thus realizing the modification from $P_\theta$ towards the desired conditional distribution $P_\theta(\cdot|\mathcal{E}_{\bar{X}_0})$.

Doob's $h$-transform is a powerful tool for adapting the generated distribution to a generic target distribution via the reweighting induced by the $h$-function. To formalize this capability, we establish the following lemma.

**Lemma 3.2.** *Let $q$ be the density function of the target distribution such that*

$$\|q/p_{\theta,0}\|_\infty \le C_q \quad \text{for a constant} \quad C_q < \infty.$$

*Let $U \sim \mathrm{Unif}(0,1)$ be independent of $\{\bar{X}_t\}_{0 \le t \le T}$. Setting $h(x_t, t) = \mathbb{P}(\mathcal{E}_{\bar{X}_0}|\bar{X}_t = x_t)$ with*

$$\mathcal{E}_{\bar{X}_0} = \left\{U \le C_q^{-1}q(\bar{X}_0)/p_{\theta,0}(\bar{X}_0)\right\}$$

*leads to $p^h_{\theta,0}(x) = q(x)$.*

The proof of Lemma 3.2 is deferred to Appendix B.1.

**Tilted Sampling Process for $P_\theta(\cdot|\mathcal{E}_{\bar{X}_0})$.** A vital advantage of Doob's $h$-transform is that sampling from the tilted distribution in (6) reduces to adding a time-dependent correction term to (4). Precisely, in order to generate samples from $P_\theta(\cdot|\mathcal{E}_{\bar{X}_0})$, we simulate the following piecewise SDE that evolves backward in time from $T$ to 0 (Rogers & Williams, 2000; Särkkä & Solin, 2019; Nguyen et al., 2025),

$$d\bar{X}^h_t = \left[-\frac{\bar{X}^h_t}{2} - s_\theta(\bar{X}^h_{t_l}, t_l) - \nabla \log h(\bar{X}^h_t, t)\right]dt + d\overline{W}_t, \quad (7)$$

where $t \in [t_{l-1}, t_l]$. We refer to $\{\bar{X}^h_t\}_{t \in [0,T]}$ as the tilted sampling process. We denote $\bar{X}^h_t \sim P^h_{\theta,t}$ as its time-$t$ marginal distribution, and $p^h_{\theta,t}$ as its marginal density. We use $\mathbb{P}^h_\theta$ to denote the corresponding path measure.

As can be seen, Doob's $h$-transform modifies the original score network $s_\theta$ by adding a dynamic Doob's correction term $\nabla \log h$. In practice, we apply a piecewise constant approximation to $\nabla \log h$, replacing $\nabla \log h(\bar{X}^h_t, t)$

by $\nabla \log h(\bar{X}^h_{t_l}, t_l)$ for $t \in [t_{l-1}, t_l]$. Consequently, we simulate the following piecewise SDE:

$$d\widehat{X}^h_t = \left[-\frac{\widehat{X}^h_t}{2} - s_\theta(\widehat{X}^h_{t_l}, t_l) - \nabla \log h(\widehat{X}^h_{t_l}, t_l)\right]dt + d\overline{W}_t, \quad (8)$$

where $t \in [t_{l-1}, t_l]$.

However, as exactly evaluating $\nabla \log h(x, t_l)$ is intractable, we must construct a reliable approximation to simulate the process $\{\widehat{X}^h_t\}_{t \in [0,T]}$.

## 4. Adaptation Algorithm Based on Doob's $h$-Transform

This section presents the DOIT (**D**oob-**O**riented **I**nference-time **T**ransformation) method and its prototypical full-rollout algorithm DOIT-Proto, solving the major challenge of the intractability of exactly computing $\nabla \log h$. We develop a simulation-based approximation to $\nabla \log h$, which leads to DOIT-Proto in Algorithm 1.

To derive a practical approximation of $\nabla \log h$, we write $\nabla \log h = \nabla h/h$, and then we approximate the numerator $\nabla h$ and the denominator $h$ separately. By Definition 3.1, the $h$-function can be straightforwardly approximated by Monte Carlo (MC) samples. Therefore, we focus on the more intricate term $\nabla h$. The following lemma expresses $\nabla h$ as a single expectation that is suitable for approximation.

**Lemma 4.1.** *Fix a discretization index $l \in \{1, \dots, L\}$ and any $x$. Assume $s_\theta(x, t)$ is continuously differentiable with respect to $x$. Then it holds that*

$$\nabla h(x, t_l) = \mathbb{E}\left[h(\bar{X}_0, 0)\nabla_{\bar{X}_{t_l}} \log \phi_\theta(\bar{X}_{t_{l-1}}|\bar{X}_{t_l})\Big| \bar{X}_{t_l} = x\right],$$

*where $\phi_\theta$ is the Gaussian density defined in (5), and the expectation is taken over the conditional distribution of the backward trajectory $(\bar{X}_{t_{l-1}}, \dots, \bar{X}_0)|\bar{X}_{t_l} = x$.*

The proof of Lemma 4.1 is deferred to Appendix B.2.

Lemma 4.1 suggests using a sample average to approximate the expectation and further approximate $\nabla h$. More importantly, since the gradient is only taken over the Gaussian transition density $\phi_\theta$, approximating $\nabla h$ does not require differentiability in the reward function.

**Sample Average Approximation to $\nabla h$ and $h$.** At time $t_l$, given a state $x_{t_l}$, we simulate $M$ trajectories $\{x^{(m)}_{t_{l-1}}, \dots, x^{(m)}_0\}_{m=1}^M$ using the transition kernel in (5). Then $\nabla h$ and $h$ are approximated respectively by

$$\nabla \widehat{h}(x_{t_l}, t_l) = \frac{1}{M}\sum_{m=1}^M h(x^{(m)}_0, 0)\nabla_{x_{t_l}} \log \phi_\theta(x^{(m)}_{t_{l-1}}|x_{t_l}),$$

$$\widehat{h}(x_{t_l}, t_l) = \frac{1}{M}\sum_{m=1}^M h(x^{(m)}_0, 0).$$

We approximate $\nabla \log h$ via a plug-in ratio approximation,

$$\nabla \log \widehat{h}(x_{t_l}, t_l) = \frac{\nabla \widehat{h}(x_{t_l}, t_l)}{\widehat{h}(x_{t_l}, t_l) \vee \eta_{t_l}}. \quad (9)$$

Here, we introduce a truncation level $\eta_{t_l} > 0$ for numerical stability. When $h(x_{t_l}, t_l)$ is small, its approximation $\widehat{h}$ will be close to zero. Naïvely using $\widehat{h}$ causes $\nabla \log \widehat{h}$ to have an exploding magnitude. We remark that such a stability issue is intrinsic to the quality of the pre-trained model, rather than a limitation of our proposed method. A small $h(x_{t_l}, t_l)$ indicates that in the pre-trained generative distribution, the desired condition is hardly satisfied by generated samples, incurring a significant gap between $P_\theta$ and $P_\theta(\cdot | \mathcal{E}_{\bar{X}_0})$.

We remark that our derived approximation of $\nabla \log h$ is different from existing approaches (Bansal et al., 2023; Nguyen et al., 2025). They often pass through the expectation to approximate

$$\mathbb{E}[h(\bar{X}_0, 0) | \bar{X}_{t_l} = x_{t_l}] \approx h(\mathbb{E}[\bar{X}_0 | \bar{X}_{t_l} = x_{t_l}], 0).$$

When taking derivatives with respect to $x_{t_l}$, such an approximation inevitably requires the differentiability of $h(\cdot, 0)$—consequently the differentiability of the reward function. In contrast, our approximation unrolls the transition from $\bar{X}_{t_l}$ to $\bar{X}_0$ into a series of incremental Gaussian transitions, i.e., $\phi_\theta$. This allows us to pass through the differentiation directly to the transition probability instead of requiring differentiability of the reward function.

Given $\nabla \log \widehat{h}$, we simulate samples using (8), achieving the adaptation from $P_\theta$ to $P_\theta(\cdot | \mathcal{E}_{\bar{X}_0})$. A prototypical algorithm is summarized below.

---

**Algorithm 1** `DOIT-Proto`

---

**Input:** Pre-trained score $s_\theta(x, t)$; the number of MC samples $M$; truncation level $\{\eta_{t_l}\}_{l=1}^L$.

1: Sample $\widehat{X}_{t_L}^h \sim \mathcal{N}(0, I)$.
2: **for** $l = L, L-1, \ldots, 1$ **do**
3:     Compute $\nabla \log \widehat{h}(\widehat{X}_{t_l}^h, t_l)$ in (9) using $M$ backward rollouts initiated at $\widehat{X}_{t_l}^h$ by simulating (4).
4:     $\nabla \log \widehat{p}_{\theta, t_l}^h(\widehat{X}_{t_l}^h) \leftarrow s_\theta(\widehat{X}_{t_l}^h, t_l) + \nabla \log \widehat{h}(\widehat{X}_{t_l}^h, t_l)$.
5:     Sample $\widehat{X}_{t_{l-1}}^h \sim \mathcal{N}(\mu_{t_l}(\widehat{X}_{t_l}^h, \nabla \log \widehat{p}_{\theta, t_l}^h), \sigma_{t_l}^2 I)$.
6: **end for**
7: **output** $\widehat{X}_0^h \sim \widehat{P}_\theta^h$.

---

Here $\widehat{P}_\theta^h$ denotes the distribution of the generated sample $\widehat{X}_0^h$. Each iteration in Algorithm 1 relies on simulating $M$ independent backward sampling trajectories using (4), which is computationally expensive. To mitigate the computational overhead, we significantly reduce the simulation cost by predicting clean data directly using the score network, while skipping all intermediate states in sampling trajectories. Algorithm 2 in Section 6.1 summarizes such computational modifications and demonstrates strong performance over state-of-the-art baselines in offline RL benchmarks.

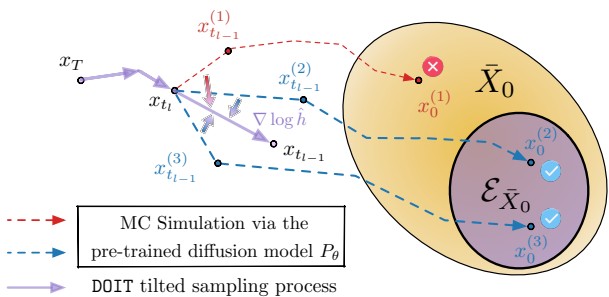

*Figure 1.* `DOIT`: At each $t_l$, we simulate $M$ trajectories (here, $M = 3$) starting from $x_{t_l}$ to approximate $\nabla \log h(x_{t_l}, t_l)$ via (9), then utilize it to modify the sampling dynamics.

## 5. Convergence Guarantee of `DOIT-Proto`

In this section, we provide a convergence guarantee of $\widehat{P}_\theta^h$ generated by `DOIT-Proto` relative to the ideal conditional data distribution $P_{\text{data}}(\cdot | \mathcal{E}_{X_0})$. Our analysis proceeds in two steps. We first quantify the approximation error of $\nabla \log h(x_{t_l}, t_l)$. We then propagate this approximation error through the tilted sampling dynamics in (7) to obtain an end-to-end distribution approximation bound.

To begin with, we impose the following assumption on the pre-trained generated distribution $P_\theta$ and score $s_\theta(x_{t_l}, t_l)$.

**Assumption 5.1.** There exist constants $\rho \in (0, 1]$ and $G > 0$ such that $\mathbb{P}(\mathcal{E}_{\bar{X}_0}) \geq \rho$, and for any index $l \in \{1, \ldots, L\}$, it holds that $\sup_x \|\nabla_x s_\theta(x, t_l)\|_2 \leq G$.

The first condition is a non-degeneracy assumption, it requires the target event $\mathcal{E}_{\bar{X}_0}$ to occur with at least probability $\rho$ under $P_\theta$. This lower bound is essential for the stability of the MC approximation in (9). When $\rho$ is extremely small, accurately approximating $\nabla \log h$ from a finite number of rollouts is fundamentally difficult, as $h(x_0^{(m)}, 0)$ is not close to zero for only a small fraction of trajectories.

The second condition imposes a global Lipschitz-type regularity on the pre-trained score network, which is a key ingredient for establishing concentration of the MC approximation. Such smoothness assumptions are reasonable in practice, since modern training typically incorporates regularization and stabilization techniques that implicitly control the network's Lipschitz continuity, such as weight decay (Krogh & Hertz, 1991; Loshchilov & Hutter, 2017).

The following lemma establishes a high probability bound on approximating $\nabla \log h$.

**Lemma 5.2** (Approximation bound of $\nabla \log h$). *Suppose Assumption 5.1 holds. Fix a discretization index $l \in \{1, \ldots, L\}$ and $\delta \in (0, 1)$. Then for a sufficiently large number of MC samples $M$ and $\eta_{t_l} = M^{-1/6}$, with proba-*

*bility at least $1 - \delta$, it holds that*

$$\mathbb{E}_{X_{t_l} \sim P_{\theta,t_l}^h} \left[ \|\nabla \log \widehat{h}(X_{t_l}, t_l) - \nabla \log h(X_{t_l}, t_l)\|_2^2 \right]$$
$$= \mathcal{O} \left( \frac{1}{\sigma_{t_l}^2} \frac{\log \frac{M}{\delta} \sqrt{\log(1/\delta)}}{M^{1/6}} \right).$$

The proof of Lemma 5.2 is deferred to Appendix B.3.

Lemma 5.2 gives a high-probability bound on the mean-squared error of the MC approximation for $\nabla \log h(\cdot, t_l)$. The error decays as the number of MC samples $M$ increases, with a dimension-independent rate $\mathcal{O}(M^{-1/6})$, reflecting the concentration of the MC approximation (9).

Moreover, the bound depends on $\sigma_{t_l}^2$, indicating that approximating $\nabla \log h$ becomes more challenging when $\sigma_{t_l}$ is small and $t_l$ is near zero. In this regime, the conditional transition $X_{t_{l-1}}|X_{t_l}$ is nearly deterministic, meaning the transition density $\phi_\theta(\cdot|X_{t_l})$ approaches a Dirac delta function. This results in an ill-behaved score function and causes the magnitude of $\nabla \log h(\cdot, t_l)$ to become extremely large. Consequently, even small approximation errors in the numerator are significantly amplified. The remaining factor $\log(M/\delta)\sqrt{\log(1/\delta)}$ comes from concentration over the MC randomness under a $1 - \delta$ guarantee.

We propagate the MC error bound in Lemma 5.2 through the tilted sampling process, which yields an end-to-end convergence guarantee by bounding the discrepancy between the target conditional distribution $P_{\text{data}}(\cdot|\mathcal{E}_{X_0})$ and the output distribution $\widehat{P}_\theta^h$ produced by `DOIT-Proto`. We begin by introducing an assumption that quantifies the discretization error incurred by (8).

**Assumption 5.3.** *For any $l \in \{1, \ldots, L\}$, and for any $t \in [t_{l-1}, t_l]$, we assume that the discretization error is uniformly bounded by $\varepsilon_{\text{dis}} \geq 0$,*

$$\mathbb{E}_{\mathbb{P}_\theta^h} \left[ \|\nabla \log h(\bar{X}_{t_l}^h, t_l) - \nabla \log h(\bar{X}_t^h, t)\|_2^2 \right] \leq \varepsilon_{\text{dis}}.$$

Under additional regularity conditions (e.g., Lipschitz continuity of $\nabla \log h$ in $(x, t)$ together with mild properties of the event $\mathcal{E}_{\bar{X}_0}$), such a bound can be derived explicitly; see Chen et al. (2023b) for reference.

**Theorem 5.4.** *Suppose Assumptions 5.1 and 5.3 hold and choose $\eta_t$ as in Lemma 5.2. With probability at least $1 - \delta$, it holds that*

$$\text{TV} \left( P_{\text{data}}(\cdot|\mathcal{E}_{X_0}), \widehat{P}_\theta^h \right)$$
$$\lesssim \frac{1}{\rho} \text{TV}(P_{\text{data}}, P_\theta) + \sqrt{\frac{\kappa_\sigma \log \frac{M}{\delta} \sqrt{\log(1/\delta)}}{M^{1/6}} + \varepsilon_{\text{dis}} T},$$

*where $\kappa_\sigma = \frac{T}{L} \sum_{l=1}^L \sigma_{t_l}^{-2}$.*

The proof of Theorem 5.4 is deferred to Appendix B.4.

Theorem 5.4 decomposes the discrepancy into two terms. The first term, $\frac{1}{\rho} \text{TV}(P_{\text{data}}, P_\theta)$, consists of two interpretable factors. The multiplier $1/\rho$ reflects the intrinsic difficulty of the conditional generation task. When $\mathbb{P}(\mathcal{E}_{\bar{X}_0})$ is small, the pre-trained model rarely generates the desired outcomes, thereby posing a significant challenge to achieving high rewards. The second factor, $\text{TV}(P_{\text{data}}, P_\theta)$, measures the discrepancy between the generated distribution and the ground truth data distribution. This factor corresponds to the explicit convergence rates of diffusion models established in prior literature (Block et al., 2020; Chen et al., 2023a; Wibisono et al., 2024). Notably, Oko et al. (2023) provided minimax rate guarantees. In our analysis, we treat it as an irreducible error inherited from the pre-training phase.

The second term bounds the total sampling error, combining the MC approximation error and the discretization error. The term $\varepsilon_{\text{dis}} T$ accounts for the discretization error introduced by the piecewise-constant approximation of $\nabla \log h$ in (8). Moreover, $\kappa_\sigma = \frac{T}{L} \sum_{l=1}^L \sigma_{t_l}^{-2}$ summarizes the accumulation of noise-scale effects, under the standard DDPM/VP setting (Ho et al., 2020), $\kappa_\sigma = \mathcal{O}\left(\log(T/t_1)\right)$.

Together, the theorem implies that when $\text{TV}(P_{\text{data}}, P_\theta) \to 0$, $\varepsilon_{\text{dis}} \to 0$, and $M$ increases, $\widehat{P}_\theta^h$ converges to $P_{\text{data}}(\cdot|\mathcal{E}_{X_0})$ with high probability in terms of the total variation distance.

## 6. Experiments

In this section, we first instantiate the efficient surrogate variant `DOIT-Practical` and detail our specific choice of the $h$-function (Section 6.1). Subsequently, we validate that `DOIT-Practical` effectively adapts the reward distribution of generated samples toward higher-value regions (Section 6.2). Finally, we demonstrate the performance of `DOIT-Practical` on offline reinforcement learning tasks (Section 6.3). Code is available at https://github.com/liamyzq/Doob_training_free_adaptation.

### 6.1. Practical Instantiation of `DOIT`

In `DOIT-Proto`, approximating $\nabla \log \widehat{h}(x_{t_l}, t_l)$ at each state $x_{t_l}$ necessitates $M$ full backward rollouts to obtain the terminal states $\{x_0^{(m)}\}_{m=1}^M$. This process incurs a prohibitive computational cost due to the additional Number of Function Evaluations (NFEs) of the score network $s_\theta$. To circumvent this bottleneck, we introduce an efficient surrogate $\widehat{x}_0^{(m)}$ to approximate $x_0^{(m)}$.

At state $x_{t_l}$, we first sample $M$ one-step lookahead states $\{x_{t_{l-1}}^{(m)}\}_{m=1}^M$ from the transition kernel (5). We then approximate the terminal state $x_0^{(m)}$ using Tweedie's formula (Efron, 2011), which calculates the posterior mean

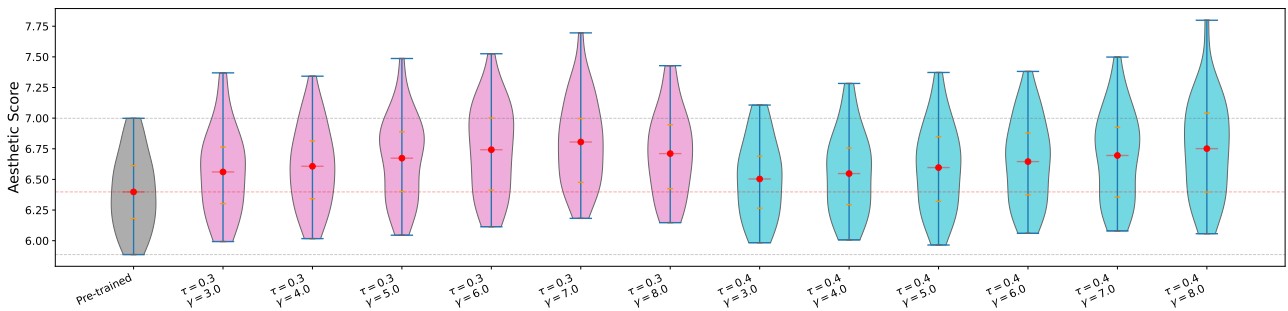

*Figure 2.* Violin plots of aesthetic scores for the samples generated by `Stable Diffusion v1.5`, comparing the vanilla generation result and applying `DOIT-Practical` across different $(\tau, \gamma)$ settings. The blue bars indicate the minimum and maximum scores, the orange bars represent the first & third quantiles, and the red marker denotes the mean.

---

**Algorithm 2** `DOIT-Practical`

---

**Input:** Pre-trained score $s_\theta(x, t)$; correction strength $\gamma$; number of MC samples $M$; time threshold $l^*$; truncation levels $\{\eta_{t_l}\}_{l=1}^L$.

1: Sample $x_{t_L}$ from $\mathcal{N}(0, I)$.
2: **for** $l = L, L-1, \ldots, 1$ **do**
3:    **if** $1 < l \leq l^*$ **then**
4:       Sample $\{x_{t_{l-1}}^{(m)}\}_{m=1}^M$ from $\mathcal{N}\left(\mu_{t_l}(x_{t_l}, s_\theta), \sigma_{t_l}^2 I\right)$.
5:       Compute $\{\widehat{x}_0^{(m)}\}_{m=1}^M$ via (10).
6:       Compute $\nabla \log \widehat{h}(x_{t_l}, t_l)$ in (9) via $\{\widehat{x}_0^{(m)}\}_{m=1}^M$.
7:       $\nabla \log \widehat{p}_{\theta, t_l}^h(x_{t_l}) \leftarrow s_\theta(x_{t_l}, t_l) + \gamma \nabla \log \widehat{h}(x_{t_l}, t_l)$.
8:       Sample $x_{t_{l-1}}$ from $\mathcal{N}(\mu_{t_l}(x_{t_l}, \nabla \log \widehat{p}_{\theta, t_l}^h), \sigma_{t_l}^2 I)$.
9:    **else**
10:      Sample $x_{t_{l-1}}$ from $\mathcal{N}\left(\mu_{t_l}(x_{t_l}, s_\theta), \sigma_{t_l}^2 I\right)$.
11:    **end if**
12: **end for**
13: **Return** $x_0$.

---

given a noisy state:

$$\mathbb{E}[x_0 \mid x_{t_{l-1}}^{(m)}] = \frac{x_{t_{l-1}}^{(m)} + (1 - e^{-t_{l-1}})s_\theta(x_{t_{l-1}}^{(m)}, t_{l-1})}{e^{-t_{l-1}/2}},$$

where the coefficients are determined by the forward SDE (1). Evaluating this expression exactly still requires calling the score network at the new lookahead states. To avoid this, we further reuse the score $s_\theta(x_{t_l}, t_l)$ computed at step $t_l$. Consequently, we define the surrogate $\widehat{x}_0^{(m)}$ as:

$$\widehat{x}_0^{(m)} = e^{t_{l-1}/2}(x_{t_{l-1}}^{(m)} + (1 - e^{-t_{l-1}})s_\theta(x_{t_l}, t_l)). \quad (10)$$

By substituting $x_0^{(m)}$ with $\widehat{x}_0^{(m)}$, `DOIT-Practical` incurs zero additional NFEs of the score network compared to the vanilla generation process. We empirically validate this approximation in Section 6.2, demonstrating that it achieves performance comparable to full trajectory simulation.

Building on this surrogate, we present `DOIT-Practical` (Algorithm 2)—an efficient implementation of `DOIT`. No-

tably, to further enhance flexibility and stability, we introduce two hyperparameters: a time threshold $l^*$ and a correction strength $\gamma$, which restrict the application of the Doob correction to specific timesteps and control its magnitude, respectively.

**Choice of $h$-function.** Practical application requires a tailored choice of the $h$-function. In reinforcement learning and energy-based modeling, the target density is typically defined via *exponential tilting* of the base distribution (Peters et al., 2010; Haarnoja et al., 2018), taking the form $q(x) \propto p_{\theta,0}(x) \exp(r(x)/\tau)$, where $\tau > 0$ is a temperature parameter controlling the sharpness of the reward distribution: smaller $\tau$ forces a stronger concentration on high-reward regions. Following this paradigm, we specify the $h$-function at the terminal time $t = 0$ as

$$h(x, 0) \propto \exp\left(r(x)/\tau\right). \quad (11)$$

This $h$-function corresponds to a particular choice of $\mathcal{E}_{\bar{X}_0}$, emphasizing high-reward regions. A detailed derivation is deferred to Appendix C.1. We run `DOIT-Practical` with the $h$-function in (11) for all subsequent experiments.

### 6.2. `DOIT` Adapts the Sampling Distribution

We demonstrate that `DOIT` adapts the generation process for each sample, effectively tilting the entire reward distribution toward higher-value regions. We conduct experiments by applying `DOIT-Practical` to `Stable Diffusion v1.5` (Rombach et al., 2022) to improve rewards of text-to-image generation.

We first use the LAION Aesthetic Score (Beaumont et al., 2022) as the reward. We generate $K = 32$ images and analyze their aesthetic score distributions. To assess performance and robustness, we sweep across the sampling temperature $\tau$ and the correction strength $\gamma$, comparing the resulting reward distributions against the pre-trained model.

The results, summarized in Figure 2, demonstrate that our method successfully transports the pre-trained reward

*Table 1.* Performance comparison on ImageReward. $K$ represents the number of particles. Combining `DOIT-Practical` with both BoK and BFS provides a significant boost. For compactness, `DOIT` denotes `DOIT-Practical` in this table.

| $K$ | BoK | BoK + `DOIT` (Ours) | FK-Steering | DAS | TreeG | SVDD | BFS | BFS + `DOIT` (Ours) |
|---|---|---|---|---|---|---|---|---|
| 4 | $0.702 \pm 0.057$ | $0.875 \pm 0.017$ | $0.743 \pm 0.037$ | $0.878 \pm 0.028$ | $0.860 \pm 0.033$ | $0.667 \pm 0.076$ | $0.882 \pm 0.029$ | $0.950 \pm 0.016$ |
| 8 | $0.896 \pm 0.031$ | $1.001 \pm 0.015$ | $0.926 \pm 0.042$ | $1.052 \pm 0.033$ | $1.023 \pm 0.018$ | $0.775 \pm 0.087$ | $1.087 \pm 0.031$ | $1.115 \pm 0.009$ |

*Table 2.* Performance comparison of different methods on D4RL locomotion tasks. Training-time methods are marked with ♣ and inference-time methods are marked with ♠. For each task, the best-performing inference-time method has its mean highlighted in bold. For compactness, `DOIT` denotes `DOIT-Practical` in this table.

| Dataset | Environment | IQL (♣) | Diffuser (♣) | D-QL (♣) | QGPO (♣) | TFG (♠) | DAS (♠) | TTS (♠) | `DOIT` (♠) (Ours) |
|---|---|---|---|---|---|---|---|---|---|
| Medium-Expert | HalfCheetah | 86.7 | 79.8 | 96.1 | 93.5 | $90.2 \pm 0.2$ | $93.3 \pm 0.3$ | $\mathbf{93.9} \pm 0.3$ | $\mathbf{93.9} \pm 0.5$ |
| Medium-Expert | Hopper | 91.5 | 107.2 | 110.7 | 108.0 | $100.2 \pm 3.5$ | $105.4 \pm 5.1$ | $104.4 \pm 3.1$ | $\mathbf{107.7} \pm 5.6$ |
| Medium-Expert | Walker2d | 109.6 | 108.4 | 109.7 | 110.7 | $108.1 \pm 0.1$ | $111.4 \pm 0.1$ | $111.4 \pm 0.1$ | $\mathbf{113.2} \pm 0.7$ |
| Medium | HalfCheetah | 47.4 | 44.2 | 50.6 | 54.1 | $53.1 \pm 0.1$ | $53.4 \pm 0.1$ | $54.8 \pm 0.1$ | $\mathbf{55.3} \pm 0.3$ |
| Medium | Hopper | 66.3 | 58.5 | 82.4 | 98.0 | $96.2 \pm 0.5$ | $71.3 \pm 2.7$ | $99.5 \pm 1.7$ | $\mathbf{101.0} \pm 0.3$ |
| Medium | Walker2d | 78.3 | 79.7 | 85.1 | 86.0 | $83.2 \pm 1.4$ | $83.9 \pm 0.9$ | $86.5 \pm 0.2$ | $\mathbf{86.9} \pm 0.3$ |
| Medium-Replay | HalfCheetah | 44.2 | 42.2 | 47.5 | 47.6 | $45.0 \pm 0.3$ | $42.2 \pm 0.1$ | $\mathbf{47.8} \pm 0.4$ | $46.9 \pm 0.3$ |
| Medium-Replay | Hopper | 94.7 | 96.8 | 100.7 | 96.9 | $93.1 \pm 0.1$ | $96.7 \pm 3.0$ | $97.4 \pm 4.0$ | $\mathbf{101.8} \pm 0.2$ |
| Medium-Replay | Walker2d | 73.9 | 61.2 | 94.3 | 84.4 | $69.8 \pm 4.0$ | $63.8 \pm 2.0$ | $79.3 \pm 9.7$ | $\mathbf{82.0} \pm 8.9$ |
| **Average (Locomotion)** | | 76.9 | 75.3 | 86.3 | 86.6 | 82.1 | 80.2 | 86.1 | **87.6** |

distribution to high-reward counterparts; some of the example images are shown in Figure 3. Notably, we identify a trade-off region: lower temperatures (corresponding to sharper reward landscapes) require lower correction strengths to maintain stability, whereas higher temperatures accommodate stronger correction. Within this regime, `DOIT-Practical` robustly improves aesthetic scores. Experimental details and sensitivity ablation studies on $(\tau, \gamma)$ are provided in Appendix C.2.

**Comparison of Surrogate and Full Simulation.** We compare the performance of the pre-trained model against `DOIT` (with $\tau = 0.3, \gamma = 6.0$) implemented with either `DOIT-Practical` or `DOIT-Proto` to approximate the correction term. Table 3 shows that `DOIT-Practical` achieves reward improvements comparable to `DOIT-Proto`. Crucially, `DOIT-Practical` requires significantly lower runtime than `DOIT-Proto`.

**Integration with Resampling Based Methods.** As discussed in Section 2, a distinct line of research focuses on reweighting and resampling strategies. Intuitively, rather than locally correcting the trajectory as in `DOIT`, these methods start with multiple candidates and evaluate them at intermediate steps, subsequently reweighting and resampling to favor higher-reward ones. Because these approaches operate on a population level while `DOIT` improves the per-sample reward, they are orthogonal and complementary.

To demonstrate the effect of this integration, we combine `DOIT-Practical` with two representative strategies: the fundamental Best-of-K (BoK) baseline and the state-of-the-art method BFS (Zhang et al., 2025b). Following the experimental setup in Zhang et al. (2025b), we use ImageRe-

ward (Xu et al., 2023) as the reward feedback and compare against several recent reweighting-based baselines, including FK-Steering (Singhal et al., 2025), DAS (Kim et al., 2025), TreeG (Guo et al., 2026a), and SVDD (Li et al., 2024). Experimental details are in Appendix C.3.

The results are presented in Table 1. We observe that combining `DOIT-Practical` with both BoK and BFS yields significant performance gains. Because `DOIT` tilts the sampling distribution toward high-reward regions, the candidate pools generated for BoK and BFS contain higher-quality samples, thereby improving the final outcome.

*Table 3.* Comparison of aesthetic score statistics between `DOIT-Practical` and `DOIT-Proto`. Runtime (seconds) is reported as the generation time per image, excluding reward evaluation time.

| | Min | Mean | Max | Runtime |
|---|---|---|---|---|
| Pre-trained | $5.737 \pm 0.102$ | $6.372 \pm 0.031$ | $7.052 \pm 0.085$ | $1.260 \pm 0.046$ |
| `DOIT-Practical` | $5.987 \pm 0.180$ | $6.726 \pm 0.072$ | $7.501 \pm 0.055$ | $1.712 \pm 0.036$ |
| `DOIT-Proto` | $6.006 \pm 0.099$ | $6.714 \pm 0.055$ | $7.553 \pm 0.113$ | $39.584 \pm 0.142$ |

### 6.3. Performance Evaluation on Offline RL Benchmarks

Diffusion policies, utilizing diffusion models as action generators, have recently emerged as a powerful paradigm in robotics (Chi et al., 2025). However, because they are typically trained on offline datasets, adapting them to inference-time objectives remains a challenge. In this section, we apply `DOIT-Practical` to the offline reinforcement learning setting, adapting diffusion policies toward higher inference-time returns to demonstrate the effectiveness of our method. We follow the setup of Lu et al. (2023), utilizing their pre-trained diffusion policies on the D4RL benchmark (Fu et al., 2020) and employing their Q-functions as reward feedback to guide the generation of

action sequences. We evaluate performance on standard D4RL locomotion tasks, and compare `DOIT-Practical` against training-based baselines, including IQL (Kostrikov et al., 2022), Diffuser (Janner et al., 2022), D-QL (Wang et al., 2022), and QGPO (Lu et al., 2023), as well as recent inference-time methods such as TFG (Ye et al., 2024), DAS (Kim et al., 2025), and TTS (Zhang et al., 2025b). Implementation details are in Appendix C.4.

The results, summarized in Table 2, demonstrate the effectiveness of our approach. `DOIT-Practical` achieves the best performance on 8 out of 9 tasks among inference-time adaptation methods, attaining an average score of **87.6**. Notably, it surpasses the previous state-of-the-art inference-time method (TTS) and competitive training-based baselines, confirming that our efficient adaptation strategy can unlock superior policy performance without requiring additional training.

## 7. Conclusion and Limitations

In this paper, we introduce `DOIT`, a training-free inference-time adaptation method for steering pre-trained diffusion models toward high-reward outcomes under generic, potentially non-differentiable reward functions. Our key perspective is to formulate the problem as transporting the pre-trained generative distribution to a reward-induced target distribution. We realize this transport via Doob's $h$-transform, which induces an additive, time-dependent correction to the sampling dynamics while keeping the pre-trained score model frozen. Based on this method, we develop two algorithmic variants: `DOIT-Proto`, a full-rollout algorithm for which we establish an end-to-end high-probability convergence guarantee, and `DOIT-Practical`, an efficient surrogate implementation used in experiments. Empirically, we demonstrate that `DOIT-Practical` reliably concentrates samples in high-reward regions and achieves strong performance on offline RL benchmarks with competitive sampling efficiency.

One potential limitation of `DOIT` is its sensitivity to the probability of the high-reward event $\mathcal{E}_{\bar{X}_0}$. Extremely rare high-reward regions can lead to large variance in estimating the Doob correction and reduced sampling efficiency. We hope future work will address this limitation.

## Impact Statement

This paper aims to contribute to progress in Machine Learning. While our work may have broader societal impacts, we do not identify any that require specific discussion at this time.

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

# A. Specific Forms of Backward Kernels

In this section, we provide the explicit forms of the Gaussian transition kernels $p_\theta(x_{t_{l-1}}|x_{t_l})$ defined in Eq. (5) for the sampling algorithms used in our experiments. We utilize the time discretization grid $0 = t_0 < t_1 < \cdots < t_L = T$, where the reverse process moves from $t_l$ to $t_{l-1}$.

Throughout this section, we relate the model's noise prediction $\epsilon_\theta(x_t, t)$ to the parameterized score function $s_\theta(x, t)$ via:

$$\epsilon_\theta(x_t, t) = -\sqrt{1 - \alpha_t}\, s_\theta(x_t, t).$$

## A.1. DDIM Sampler

The generalized Denoising Diffusion Implicit Model (DDIM) (Song et al., 2020a) sampler updates the state based on the noise schedule $\alpha_t$ and a stochasticity hyperparameter $\eta \in [0, 1]$. When $\eta = 0$, the process is deterministic; when $\eta = 1$, it recovers the original DDPM (Ho et al., 2020) sampling process.

Given the noisy sample $x_{t_l}$ at time step $t_l$, we first estimate the clean data $\widehat{x}_0$ using the model prediction $\epsilon_\theta(x_{t_l}, t_l)$:

$$\widehat{x}_0(x_{t_l}, t_l) = \frac{x_{t_l} - \sqrt{1 - \alpha_{t_l}}\, \epsilon_\theta(x_{t_l}, t_l)}{\sqrt{\alpha_{t_l}}}.$$

We define the variance of the transition kernel as:

$$\sigma_{t_l}^2(\eta) = \eta^2 \left( \frac{1 - \alpha_{t_{l-1}}}{1 - \alpha_{t_l}} \right) \left( 1 - \frac{\alpha_{t_l}}{\alpha_{t_{l-1}}} \right).$$

The update rule for the subsequent sample $x_{t_{l-1}}$ combines the estimated clean data, the direction pointing to $x_{t_l}$, and random noise:

$$x_{t_{l-1}} = \sqrt{\alpha_{t_{l-1}}}\, \widehat{x}_0(x_{t_l}, t_l) + \sqrt{1 - \alpha_{t_{l-1}} - \sigma_{t_l}^2(\eta)}\, \epsilon_\theta(x_{t_l}, t_l) + \sigma_{t_l}(\eta)\, Z,$$

where $Z \sim \mathcal{N}(0, I)$. Substituting $\widehat{x}_0$ back into the update rule yields the transition mean $\mu_\theta$. The kernel is given by:

$$p_\theta(x_{t_{l-1}}|x_{t_l}) = \mathcal{N}\left( \mu_{t_l}(x_{t_l}, s_\theta), \sigma_{t_l}^2(\eta)I \right),$$

where the mean is:

$$\mu_{t_l}(x_{t_l}, s_\theta) = \sqrt{\frac{\alpha_{t_{l-1}}}{\alpha_{t_l}}}\, x_{t_l} + \left( \sqrt{1 - \alpha_{t_{l-1}} - \sigma_{t_l}^2(\eta)} - \sqrt{1 - \alpha_{t_l}} \sqrt{\frac{\alpha_{t_{l-1}}}{\alpha_{t_l}}} \right) \epsilon_\theta(x_{t_l}, t_l).$$

## A.2. Euler Ancestral SDE

For the Euler ancestral sampler (Karras et al., 2022), we adopt the EDM noise parameterization. We first convert the noise schedule values $\alpha_{t_l}$ and $\alpha_{t_{l-1}}$ into the EDM noise standard deviations:

$$\widetilde{\sigma}_{t_l} = \sqrt{\frac{1 - \alpha_{t_l}}{\alpha_{t_l}}}, \qquad \widetilde{\sigma}_{t_{l-1}} = \sqrt{\frac{1 - \alpha_{t_{l-1}}}{\alpha_{t_{l-1}}}}.$$

The ancestral SDE step splits the transition from $\widetilde{\sigma}_{t_l}$ to $\widetilde{\sigma}_{t_{l-1}}$ into a deterministic "down" step and a stochastic "up" step. The variances for these components are defined as:

$$\widetilde{\sigma}_{t_l,\text{up}}^2 = \frac{\widetilde{\sigma}_{t_{l-1}}^2}{\widetilde{\sigma}_{t_l}^2} \left( \widetilde{\sigma}_{t_l}^2 - \widetilde{\sigma}_{t_{l-1}}^2 \right),$$

$$\widetilde{\sigma}_{t_l,\text{down}}^2 = \widetilde{\sigma}_{t_{l-1}}^2 - \widetilde{\sigma}_{t_l,\text{up}}^2.$$

We define the EDM drift direction $d_{t_l}$ directly via the noise prediction:

$$d_{t_l} = \epsilon_\theta(x_{t_l}, t_l).$$

A single Euler ancestral SDE step from $t_l$ to $t_{l-1}$ takes the form:

$$x_{t_{l-1}} = x_{t_l} + d_{t_l} \left( \widetilde{\sigma}_{t_l,\text{down}} - \widetilde{\sigma}_{t_l} \right) + \widetilde{\sigma}_{t_l,\text{up}} Z, \qquad Z \sim \mathcal{N}(0, I).$$

Therefore, the transition kernel matches the Gaussian form in Eq. (5):

$$p_\theta(x_{t_{l-1}}|x_{t_l}) = \mathcal{N}\left( \mu_{t_l}(x_{t_l}, s_\theta), \sigma_{t_l}^2 I \right),$$

with the mean and variance given by:

$$\mu_{t_l}(x_{t_l}, s_\theta) = x_{t_l} + \epsilon_\theta(x_{t_l}, t_l) \left( \widetilde{\sigma}_{t_l,\text{down}} - \widetilde{\sigma}_{t_l} \right),$$

$$\sigma_{t_l}^2 = \widetilde{\sigma}_{t_l,\text{up}}^2.$$

## B. Proofs

### B.1. Proof of Lemma 3.2

*Proof.* According to the definition of the tilted density at time $t$ in (6), taking $t = 0$ yields:

$$p_{\theta,0}^h(x) = \frac{h(x,0)p_{\theta,0}(x)}{\mathbb{P}(\mathcal{E}_{\bar{X}_0})}. \tag{12}$$

We first compute the term $h(x,0)$. By the definition of the $h$-function and the event $\mathcal{E}_{\bar{X}_0}$, we have:

$$h(x,0) = \mathbb{P}(\mathcal{E}_{\bar{X}_0} \mid \bar{X}_0 = x)$$
$$= \mathbb{P}\left(U \leq \frac{q(\bar{X}_0)}{C_q \cdot p_{\theta,0}(\bar{X}_0)} \middle| \bar{X}_0 = x\right).$$

Since the uniform random variable $U$ is independent of the generated sample $\bar{X}_0$, the conditional probability simplifies to:

$$h(x,0) = \mathbb{P}\left(U \leq \frac{q(x)}{C_q \cdot p_{\theta,0}(x)}\right).$$

Given that $U \sim \text{Unif}(0,1)$ and the assumption $\|q/p_{\theta,0}\|_\infty \leq C_q$ implies $0 \leq \frac{q(x)}{C_q p_{\theta,0}(x)} \leq 1$, the probability is exactly the value of the threshold,

$$h(x,0) = \frac{q(x)}{C_q \cdot p_{\theta,0}(x)}. \tag{13}$$

Next, we compute the marginal probability of the event $\mathbb{P}(\mathcal{E}_{\bar{X}_0})$,

$$\mathbb{P}(\mathcal{E}_{\bar{X}_0}) = \int h(x,0)p_{\theta,0}(x) = \int \frac{q(x)}{C_q \cdot p_{\theta,0}(x)}p_{\theta,0}(x)\mathrm{d}x = \frac{1}{C_q}. \tag{14}$$

Finally, substituting (13) and (14) back into (12), we obtain,

$$p_{\theta,0}^h(x) = \frac{\frac{q(x)}{C_q \cdot p_{\theta,0}(x)} \cdot p_{\theta,0}(x)}{1/C_q} = q(x).$$

This completes the proof. $\square$

### B.2. Proof of Lemma 4.1

*Proof.* Recalling the definition of the $h$-function in Definition 3.1, we can write $\nabla_x h(x, t_l)$ as,

$$\nabla_x h(x, t_l) = \nabla_x \mathbb{P}(\mathcal{E}_{\bar{X}_0}|\bar{X}_{t_l} = x)$$
$$= \nabla_x \int \mathbb{P}(\mathcal{E}_{\bar{X}_0}|\bar{X}_0 = x_0)\, p_{\theta,0|t_l}(x_0|x)\mathrm{d}x_0, \tag{15}$$

where $p_{\theta,0|t_l}(\bar{X}_0|\bar{X}_{t_l})$ represents the transition density of the conditional distribution $\bar{X}_0|\bar{X}_{t_l}$. In the backward process, the transition density $p_{\theta,0|t_l}(x_0|x_{t_l})$ is induced by (5). It can be represented as:

$$p_{\theta,0|t_l}(x_0|x_{t_l}) = \int \prod_{j=1}^{l} \phi_\theta(x_{t_{j-1}}|x_{t_j})\mathrm{d}x_{t_1}\ldots\mathrm{d}x_{t_{l-1}}, \tag{16}$$

where $\phi_\theta$ is the Gaussian density defined in (5). Plugging (16) into (15), we get,

$$\nabla_x \int \mathbb{P}(\mathcal{E}_{\bar{X}_0}|\bar{X}_0 = x_0)\, p_{\theta,0|t_l}(x_0|x)\mathrm{d}x_0$$
$$= \nabla_x \int \mathbb{P}(\mathcal{E}_{\bar{X}_0}|\bar{X}_0 = x_0)\left(\int \prod_{j=1}^{l} \phi_\theta(x_{t_{j-1}}|x_{t_j})\mathrm{d}x_{t_1}\ldots\mathrm{d}x_{t_{l-1}}\right)\mathrm{d}x_0$$
$$= \iint \mathbb{P}(\mathcal{E}_{\bar{X}_0}|\bar{X}_0 = x_0)\,\nabla_x\phi_\theta(x_{t_{l-1}}|x)\, p_{\theta,0|t_{l-1}}(x_0|x_{t_{l-1}})\mathrm{d}x_{t_{l-1}}\mathrm{d}x_0$$
$$= \iint h(x_0,0)\,\nabla_x \log \phi_\theta(x_{t_{l-1}}|x)\,\phi_\theta(x_{t_{l-1}}|x)p_{\theta,0|t_{l-1}}(x_0|x_{t_{l-1}})\mathrm{d}x_{t_{l-1}}\mathrm{d}x_0$$
$$= \mathbb{E}\left[h(\bar{X}_0,0)\nabla_{\bar{X}_{t_l}} \log \phi_\theta(\bar{X}_{t_{l-1}}|\bar{X}_{t_l})\middle|\bar{X}_{t_l} = x\right], \tag{17}$$

where the expectation is taken over conditional distribution of the backward trajectory $(\bar{X}_{t_{l-1}}, \ldots, \bar{X}_0)|\bar{X}_{t_l} = x$.

The second equality holds due to $\int h(x_0, 0)p_{\theta,0|t_{l-1}}(x_0|x_{t_{l-1}})\mathrm{d}x_0 \leq 1$. Furthermore, the Gaussian transition density $\phi_\theta(\cdot|x)$ is continuously differentiable with respect to $x$, and its gradient is dominated by an integrable function due to the exponential decay of the Gaussian tail. Then the interchange of differentiation and integration is justified by the Leibniz integral rule.

Equation (17) yields the desired expression, which concludes the proof. $\qquad\square$

### B.3. Proof of Lemma 5.2

*Proof.* In the beginning of the proof, it's helpful to discuss about the Monte Carlo (MC) randomness in the backward sampler. It actually comes from the Gaussian noises used in each reverse transition.

For each trajectory $m \in \{1, \ldots, M\}$, let $\{z_j^{(m)}\}_{j=1}^l$ be i.i.d. with $z_j^{(m)} \sim \mathcal{N}(0, I_d)$, independent across $j$ and $m$. By the transition kernel in (5)

$$\bar{X}_{t_{l-1}}|\bar{X}_{t_l} = x_{t_l} \sim \mathcal{N}\left(\mu_{t_l}(x_{t_l}, s_\theta), \sigma_{t_l}^2 I\right), \tag{18}$$

By the reparameterization trick, we can couple one step as

$$x_{t_{j-1}}^{(m)} = \mu_{t_j}\left(x_{t_j}^{(m)}, s_\theta\right) + \sigma_{t_j} z_j^{(m)}, \qquad j = l, l-1, \ldots, 1, \quad x_{t_l}^{(m)} = x_{t_l}.$$

Hence the whole backward path is a deterministic function of $(x_{t_l}, z_{1:l}^{(m)})$,

$$(x_{t_0}^{(m)}, \ldots, x_{t_{l-1}}^{(m)}) = \mathcal{G}_{\theta,l}\left(x_{t_l}; z_{1:l}^{(m)}\right),$$

so two MC estimators $\widehat{h}$ and $\nabla\widehat{h}$ can be written as functions of $\mathcal{G}_{\theta,l}\left(x_{t_l}; z_{1:l}^{(m)}\right)$, making the MC randomness explicit through $\{z_{1:l}^{(m)}\}_{m=1}^M$ (conditional on $x_{t_l}$).

For simplicity, denote by

$$z_{\mathrm{mc}} = \{z_j^{(m)}\}_{m=1,\ldots,M;\, j=1,\ldots,l}, \qquad z_j^{(m)} \overset{\text{i.i.d.}}{\sim} \mathcal{N}(0, I_d),$$

the Gaussian noises used to generate the $M$ Monte Carlo backward trajectories at time $t_l$ (conditional on a given input $x_{t_l}$).

We now begin the proof. To bound the discrepancy between $\nabla \log \widehat{h}(x_{t_l}, t_l)$ and $\nabla \log h(x_{t_l}, t_l)$, we write

$$||\nabla \log \widehat{h}(x_{t_l}, t_l) - \nabla \log h(x_{t_l}, t_l)||_2 \tag{19}$$

$$\leq \left\| \frac{\nabla\widehat{h}(x_{t_l}, t_l) - \nabla h(x_{t_l}, t_l)}{\widehat{h}(x_{t_l}, t_l) \vee \eta_{t_l}} \right\|_2 + \left\| \nabla h(x_{t_l}, t_l) \frac{h(x_{t_l}, t_l) - (\widehat{h}(x_{t_l}, t_l) \vee \eta_{t_l})}{(\widehat{h}(x_{t_l}, t_l) \vee \eta_{t_l})\, h(x_{t_l}, t_l)} \right\|_2$$

$$\leq \frac{\left\| \nabla\widehat{h}(x_{t_l}, t_l) - \nabla h(x_{t_l}, t_l) \right\|_2}{\eta_{t_l}} + ||\nabla \log h(x_{t_l}, t_l)||_2 \frac{\left| h(x_{t_l}, t_l) - (\widehat{h}(x_{t_l}, t_l) \vee \eta_{t_l}) \right|}{\eta_{t_l}}. \tag{20}$$

By $(a+b)^2 \leq 2a^2 + 2b^2$, (20) implies

$$\mathbb{E}_{X_{t_l} \sim P_{\theta,t_l}^h}\left[ ||\nabla \log \widehat{h}(X_{t_l}, t_l) - \nabla \log h(X_{t_l}, t_l)||_2^2 \right]$$

$$\leq \underbrace{\mathbb{E}_{X_{t_l} \sim P_{\theta,t_l}^h}\left[ \frac{2\left\| \nabla\widehat{h}(X_{t_l}, t_l) - \nabla h(X_{t_l}, t_l) \right\|_2^2}{\eta_{t_l}^2} \right]}_{\mathcal{H}_1}$$

$$+ \underbrace{\mathbb{E}_{X_{t_l} \sim P_{\theta,t_l}^h}\left[ 2||\nabla \log h(X_{t_l}, t_l)||_2 \frac{\left| h(X_{t_l}, t_l) - (\widehat{h}(X_{t_l}, t_l) \vee \eta_{t_l}) \right|^2}{\eta_{t_l}^2} \right]}_{\mathcal{H}_{1,2}}. \tag{21}$$

For the second term $\mathcal{H}_{1,2}$, we have

$$\mathbb{E}_{X_{t_l}\sim P^h_{\theta,t_l}}\left[2||\nabla\log h(X_{t_l},t_l)||_2\frac{\left|h(X_{t_l},t_l)-(\widehat{h}(X_{t_l},t_l)\vee\eta_{t_l})\right|^2}{\eta_{t_l}^2}\right]$$

$$\leq\mathbb{E}_{X_{t_l}\sim P^h_{\theta,t_l}}\left[4\left\|\nabla_{X_{t_l}}\log h(X_{t_l},t_l)\right\|_2^2\frac{((h(X_{t_l},t_l)\vee\eta_{t_l})-(\widehat{h}(X_{t_l},t_l)\vee\eta_{t_l}))^2}{\eta_{t_l}^2}\right]$$

$$+\mathbb{E}_{X_{t_l}\sim P^h_{\theta,t_l}}\left[4\left\|\nabla_{X_{t_l}}\log h(X_{t_l},t_l)\right\|_2^2\frac{(h(X_{t_l},t_l)-(h(X_{t_l},t_l)\vee\eta_{t_l}))^2}{\eta_{t_l}^2}\right]$$

$$\leq\underbrace{\mathbb{E}_{X_{t_l}\sim P^h_{\theta,t_l}}\left[4\left\|\nabla_{X_{t_l}}\log h(X_{t_l},t_l)\right\|_2^2\frac{(h(X_{t_l},t_l)-\widehat{h}(X_{t_l},t_l))^2}{\eta_{t_l}^2}\right]}_{\bar{\mathcal{H}}_2}$$

$$+\underbrace{\mathbb{E}_{X_{t_l}\sim P^h_{\theta,t_l}}\left[4\left\|\nabla_{X_{t_l}}\log h(X_{t_l},t_l)\right\|_2^2\frac{(h(X_{t_l},t_l)-(h(X_{t_l},t_l)\vee\eta_{t_l}))^2}{\eta_{t_l}^2}\right]}_{\mathcal{H}_3}. \tag{22}$$

The second inequality holds because $f(x)=x\vee\eta_{t_l}=\max(x,\eta_{t_l})$ is 1-Lipschitz. We derive an upper bound for $\bar{\mathcal{H}}_2$ using the following lemma.

**Lemma B.1.** *Fix a discretization index $l\in\{1,\ldots,L\}$. The following inequality holds,*

$$\|\nabla_{x_{t_l}}\log h(x_{t_l},t_l)\|_2\leq\frac{\|\nabla_{x_{t_l}}\mu_{t_l}(x_{t_l},s_\theta)\|_2}{\sigma_{t_l}}\sqrt{2\log\frac{1}{h(x_{t_l},t_l)}}.$$

The proof of Lemma B.1 is deferred to Appendix B.5.

Then we are ready to bound $\bar{\mathcal{H}}_2$

$$\bar{\mathcal{H}}_2=\int\left[4\left\|\nabla_{x_{t_l}}\log h(x_{t_l},t_l)\right\|_2^2\frac{(h(x_{t_l},t_l)-\widehat{h}(x_{t_l},t_l))^2}{\eta_{t_l}^2}\right]p^h_{\theta,t_l}(x_{t_l})\mathrm{d}x_{t_l}.$$

By (28), Lemma B.1 and Assumption 5.1 we have

$$\bar{\mathcal{H}}_2\leq\int\frac{8\sup_{x_{t_l}}\|\nabla_{x_{t_l}}\mu_{t_l}(x_{t_l},s_\theta)\|_2^2}{\sigma_{t_l}^2\eta_{t_l}^2}(h(x_{t_l},t_l)-\widehat{h}(x_{t_l},t_l))^2\log\frac{1}{h(x_{t_l},t_l)}\frac{p_{\theta,t_l}(x_{t_l})h(x_{t_l},t_l)}{\mathbb{P}(\mathcal{E}_{\bar{X}_0})}\mathrm{d}x_{t_l}$$

$$\leq\mathbb{E}_{X_{t_l}\sim P_{\theta,t_l}}\left[\frac{8C_G^2(h(X_{t_l},t_l)-\widehat{h}(X_{t_l},t_l))^2}{e\sigma_{t_l}^2\eta_{t_l}^2\rho}\right]$$

$$\leq\underbrace{\mathbb{E}_{X_{t_l}\sim P_{\theta,t_l}}\left[\frac{4C_G^2(h(X_{t_l},t_l)-\widehat{h}(X_{t_l},t_l))^2}{\sigma_{t_l}^2\eta_{t_l}^2\rho}\right]}_{\mathcal{H}_2}, \tag{23}$$

where we invoke $\sup_{x_{t_l}}\|\nabla_{x_{t_l}}\mu_{t_l}(x_{t_l},s_\theta)\|_2\leq C_G$. Note that $C_G$ is a constant determined solely by the scheduler and $G$, which follows from the fact that $\mu_{t_l}$ is a linear combination of $x_{t_l}$ and $s_\theta$ with uniformly bounded coefficients, and $\sup_x\|\nabla_x s_\theta(x,t_l)\|_2\leq G$ by Assumption 5.1.

The second inequality holds due to $\log\frac{1}{h(x_t,t)}h(x_t,t)\leq\frac{1}{e}$ and $h(x_t,t)\in(0,1]$. Then we can conclude

$$\mathbb{E}_{X_t\sim P^h_{\theta,t_l}}\left[||\nabla\log\widehat{h}(X_{t_l},t_l)-\nabla\log h(X_{t_l},t_l)||_2^2\right]\leq\mathcal{H}_1+\mathcal{H}_2+\mathcal{H}_3. \tag{24}$$

Next, we bound $\mathbb{E}_{z_{\mathrm{mc}}}[\mathcal{H}_1]$ and $\mathbb{E}_{z_{\mathrm{mc}}}[\mathcal{H}_2]$, and then obtain high-probability bounds for $|\mathcal{H}_1-\mathbb{E}_{z_{\mathrm{mc}}}\mathcal{H}_1|$ and $|\mathcal{H}_2-\mathbb{E}_{z_{\mathrm{mc}}}\mathcal{H}_2|$ via concentration over the independent noises $z_{\mathrm{mc}}$. In the end, we bound $\mathcal{H}_3$. Combining them together, we can derive a high probability bound for the approximation for $\nabla\log h$ in (24).

**Bounding** $\mathbb{E}_{z_{\mathrm{mc}}}[\mathcal{H}_1]$   For $\nabla\widehat{h}(x_{t_l}, t_l)$, we define $H_i = h(x_0^{(i)}, 0)\frac{1}{\sigma_{t_l}^2}\nabla_{x_{t_l}}(\mu_{t_l}(x_{t_l}, s_\theta))(x_{t_{l-1}}^{(i)} - \mu_{t_l}(x_{t_l}, s_\theta))$.

By definition, we have $\nabla\widehat{h}(x_t, t) = \sum_{i=1}^M \frac{1}{M}H_i$, then by Lemma 4.1, we immediately have

$$\mathbb{E}_{z_{\mathrm{mc}}}[\nabla\widehat{h}(x_{t_l}, t_l)] = \mathbb{E}_{(x_0,\ldots,x_{t_{l-1}})|x_{t_l}}[\nabla\widehat{h}(x_{t_l}, t_l)]$$
$$= \nabla h(x_{t_l}, t_l).$$

We want to prove it is a sub-Gaussian random vector for a fixed $x_{t_l}$.

Firstly note that $x_{t_{l-1}}^{(i)} \sim \mathcal{N}(\mu_{t_l}(x_{t_l}, s_\theta), \sigma_{t_l}^2 I)$, then we have

$$\frac{1}{\sigma_{t_l}^2}\nabla_{x_{t_l}}(\mu_{t_l}(x_{t_l}, s_\theta))(x_{t_{l-1}}^{(i)} - \mu_{t_l}(x_{t_l}, s_\theta)) \sim \mathcal{N}(0, \sigma_{t_l}^{-2}(\nabla_{x_{t_l}}(\mu_{t_l}(x_{t_l}, s_\theta))(\nabla_{x_{t_l}}(\mu_{t_l}(x_{t_l}, s_\theta))^\top))).$$

Then for $\forall v \in \mathbb{R}^d, ||v||_2 = 1$, we have

$$v^\top(\sigma_{t_l}^{-2}\nabla_{x_{t_l}}(\mu_{t_l}(x_{t_l}, s_\theta))(x_{t_{l-1}}^{(i)} - \mu_{t_l}(x_{t_l}, s_\theta)) \sim \mathcal{N}(0, \sigma_{t_l}^{-2}||(\nabla_{x_{t_l}}(\mu_{t_l}(x_{t_l}, s_\theta))v||_2^2),$$

which implies

$$P(|v^\top(\sigma_{t_l}^{-2}\nabla_{x_{t_l}}(\mu_{t_l}(x_{t_l}, s_\theta))(x_{t_{l-1}}^{(i)} - \mu_{t_l}(x_{t_l}, s_\theta))| \geq t)$$
$$\leq 2\exp\left(-\frac{t^2}{2\sigma_{t_l}^{-2}||(\nabla_{x_{t_l}}(\mu_{t_l}(x_{t_l}, s_\theta)))v||_2^2}\right)$$
$$\leq 2\exp\left(-\frac{t^2}{2\sigma_{t_l}^{-2}\left\|\nabla_{x_{t_l}}(\mu_{t_l}(x_{t_l}, s_\theta))\right\|_2^2}\right). \tag{25}$$

Notice that $|v^\top H_i| \leq |v^\top(\sigma_{t_l}^{-2}\nabla_{x_{t_l}}(\mu_{t_l}(x_{t_l}, s_\theta))(x_{t_{l-1}}^{(i)} - \mu_{t_l}(x_{t_l}, s_\theta))|$, by (25)

$$P(|v^\top H_i| \geq t) \leq P(|v^\top(\sigma_{t_l}^{-2}\nabla_{x_{t_l}}(\mu_{t_l}(x_{t_l}, s_\theta))(x_{t_{l-1}}^{(i)} - \mu_{t_l}(x_{t_l}, s_\theta))| \geq t)$$
$$\leq 2\exp\left(-\frac{t^2}{2\sigma_{t_l}^{-2}\left\|\nabla_{x_{t_l}}(\mu_{t_l}(x_{t_l}, s_\theta))\right\|_2^2}\right).$$

By Lemma B.1, we have

$$\left|v^\top\nabla h(x_{t_l}, t_l)\right| \leq ||\nabla h(x_{t_l}, t_l)||_2$$
$$\leq h(x_{t_l}, t_l)||\nabla\log h(x_{t_l}, t_l)||_2$$
$$\leq \frac{||\nabla_{x_{t_l}}(\mu_{t_l}(x_{t_l}, s_\theta))||_2}{\sigma_{t_l}}\sqrt{2(h(x_{t_l}, t_l))^2\log\frac{1}{h(x_{t_l}, t_l)}}$$
$$\leq \frac{||\nabla_{x_{t_l}}(\mu_{t_l}(x_{t_l}, s_\theta))||_2}{\sigma_{t_l}}. \tag{26}$$

It also directly implies $||\nabla h(x_t, t)||_2 \leq \sigma_{t_l}^{-1}||\nabla_{x_{t_l}}(\mu_{t_l}(x_{t_l}, s_\theta))||_2 \leq \sigma_{t_l}^{-1}C_G$.

Then

$$P(|v^\top H_i - v^\top\nabla h(x_{t_l}, t_l)| \geq t)$$
$$\leq P(|v^\top H_i| + |v^\top\nabla h(x_{t_l}, t_l)| \geq t)$$
$$\leq P\left(|v^\top H_i| \geq t - \sigma_{t_l}^{-1}||\nabla_{x_{t_l}}(\mu_{t_l}(x_{t_l}, s_\theta))||_2\right)$$
$$\leq 2\exp\left(-\frac{\max\left(t - \sigma_{t_l}^{-1}||\nabla_{x_{t_l}}(\mu_{t_l}(x_{t_l}, s_\theta))||_2, 0\right)^2}{2\sigma_{t_l}^{-2}||\nabla_{x_{t_l}}(\mu_{t_l}(x_{t_l}, s_\theta))||_2^2}\right). \tag{27}$$

We claim (27) can imply

$$P(|v^\top H_i - v^\top \nabla h(x_{t_l}, t_l)| \geq t) \leq 2\exp\left(-\frac{t^2}{8\sigma_{t_l}^{-2}\|\nabla_{x_{t_l}}(\mu_{t_l}(x_{t_l}, s_\theta))\|_2^2}\right).$$

To check it, first consider $t \leq 2\sigma_{t_l}^{-1}\|\nabla_{x_{t_l}}(\mu_{t_l}(x_{t_l}, s_\theta))\|_2$, and we have

$$2\exp\left(-\frac{\max\left(t - \sigma_{t_l}^{-1}\|\nabla_{x_{t_l}}(\mu_{t_l}(x_{t_l}, s_\theta))\|_2, 0\right)^2}{2\sigma_{t_l}^{-2}\|\nabla_{x_{t_l}}(\mu_{t_l}(x_{t_l}, s_\theta))\|_2^2}\right) \geq 2e^{-1/2} \geq 1,$$

and

$$2\exp\left(-\frac{t^2}{8\sigma_{t_l}^{-2}\|\nabla_{x_{t_l}}(\mu_{t_l}(x_{t_l}, s_\theta))\|_2^2}\right) \geq 2e^{-1/2} \geq 1,$$

so these two inequalities both are trivial.

When $t \geq 2\sigma_{t_l}^{-1}\|\nabla_{x_{t_l}}(\mu_{t_l}(x_{t_l}, s_\theta))\|_2$,

$$2\exp\left(-\frac{t^2}{8\sigma_{t_l}^{-2}\|\nabla_{x_{t_l}}(\mu_{t_l}(x_{t_l}, s_\theta))\|_2^2}\right) \geq 2\exp\left(-\frac{\max\left(t - \sigma_{t_l}^{-1}\|\nabla_{x_{t_l}}(\mu_{t_l}(x_{t_l}, s_\theta))\|_2, 0\right)^2}{2\sigma_{t_l}^{-2}\|\nabla_{x_{t_l}}(\mu_{t_l}(x_{t_l}, s_\theta))\|_2^2}\right),$$

because $t - \sigma_{t_l}^{-1}\|\nabla_{x_{t_l}}(\mu_{t_l}(x_{t_l}, s_\theta))\|_2 \geq \frac{t}{2}$. Then we can conclude

$$P(|v^\top H_i - v^\top \nabla h(x_{t_l}, t_l)| \geq t) \leq 2\exp\left(-\frac{t^2}{8\sigma_{t_l}^{-2}\|\nabla_{x_{t_l}}(\mu_{t_l}(x_{t_l}, s_\theta))\|_2^2}\right).$$

Then it implies for any $k \in \{1, ..., d\}$, let $e_k = (0, \ldots, 0, 1, 0, \ldots, 0)^\top \in \mathbb{R}^d$ with the 1 in the $k$-th coordinate, it holds that

$$\begin{aligned}
\mathrm{Var}\left(e_k^\top(\nabla\widehat{h}(x_{t_l}, t_l) - \nabla h(x_{t_l}, t_l))\right) &= \mathrm{Var}\left(\frac{1}{M}\sum_{i=1}^{M} e_k^\top(H_i - \nabla h(x_{t_l}, t_l))\right) \\
&\leq \frac{1}{M}\mathrm{Var}(e_k^\top(H_i - \nabla h(x_{t_l}, t_l))) \\
&\leq \frac{4\|\nabla_{x_{t_l}}(\mu_{t_l}(x_{t_l}, s_\theta))\|_2^2}{\sigma_{t_l}^2 M}.
\end{aligned}$$

This leads to the upper bound of $\mathbb{E}_{z_{\mathrm{mc}}}[\mathcal{H}_1]$

$$\begin{aligned}
\mathbb{E}_{z_{\mathrm{mc}}}[\mathcal{H}_1] &= \int \mathbb{E}_{z_{\mathrm{mc}}}\left[\frac{2\left\|\nabla\widehat{h}(x_{t_l}, t_l) - \nabla h(x_{t_l}, t_l)\right\|_2^2}{\eta_{t_l}^2}\right] p_{\theta, t_l}^h(x_{t_l})\mathrm{d}x_{t_l} \\
&= \int \mathbb{E}_{z_{\mathrm{mc}}}\left[\frac{2\sum_{k=1}^{d}\left(e_k^\top(\nabla\widehat{h}(x_{t_l}, t_l) - \nabla h(x_{t_l}, t_l))\right)^2}{\eta_{t_l}^2}\right] p_{\theta, t_l}^h(x_{t_l})\mathrm{d}x_{t_l} \\
&\leq \int \frac{8d\sup\|\nabla_{x_{t_l}}(\mu_{t_l}(x_{t_l}, s_\theta))\|_2^2}{\eta_{t_l}^2 \sigma_{t_l}^2 M} p_{\theta, t_l}^h(x_{t_l})\mathrm{d}x_{t_l} \\
&\leq \frac{8dC_G^2}{\eta_{t_l}^2 \sigma_{t_l}^2 M}.
\end{aligned}$$

The last inequality holds due to Assumption 5.1 and $\sup_{x_{t_l}}\|\nabla_{x_{t_l}}\mu_{t_l}(x_{t_l}, s_\theta)\|_2 \leq C_G$.

**Bounding** $\mathbb{E}_{z_{\mathrm{mc}}}[\mathcal{H}_2]$    We first prove the concentration bound for $\widehat{h}$. Since $\widehat{h}(x_{t_l}, t_l) \in [0, 1]$, apply hoeffding's inequality for bounded random variables yields, for any $\varepsilon > 0$,

$$\mathbb{P}\big(|\widehat{h}(x_{t_l}, t_l) - h(x_{t_l}, t_l)| \geq \varepsilon\big) \;\leq\; 2\exp\big(-2M\varepsilon^2\big).$$

It implies

$$\mathbb{E}_{z_{\mathrm{mc}}}[\widehat{h}(x_{t_l}, t_l) - h(x_{t_l}, t_l)]^2 \leq \frac{1}{4M}. \tag{28}$$

Then we are ready to bound $\mathbb{E}_{z_{\mathrm{mc}}}[\mathcal{H}_2]$

$$\mathbb{E}_{z_{\mathrm{mc}}}[\mathcal{H}_2] = \mathbb{E}_{z_{\mathrm{mc}}}\left[\mathbb{E}_{X_{t_l} \sim P_{\theta, t_l}}\left[\frac{4C_G^2(h(X_{t_l}, t_l) - \widehat{h}(X_{t_l}, t_l))^2}{\sigma_{t_l}^2 \eta_{t_l}^2 \rho}\right]\right].$$

By (28), we have

$$\mathbb{E}_{z_{\mathrm{mc}}}[\mathcal{H}_2] \leq \frac{C_G^2}{M\sigma_{t_l}^2 \eta_{t_l}^2 \rho}. \tag{29}$$

**Bounding** $|\mathcal{H}_1 - \mathbb{E}_{z_{\mathrm{mc}}}[\mathcal{H}_1]|$    We prove a high probability bound for $|\mathcal{H}_1 - \mathbb{E}_{z_{\mathrm{mc}}}[\mathcal{H}_1]|$.

**Lemma B.2** (High-probability bound). *For any $\delta \in (0, 1)$, letting*

$$s = \log \frac{2M}{\delta}, \qquad R^2 = d + 2\sqrt{ds} + 2s,$$

*we have $\mathbb{P}(\Omega_R^c) \leq \delta/2$ for the event*

$$\Omega_R = \Big\{\max_{1 \leq i \leq M} \|z_l^{(i)}\|_2 \leq R\Big\}.$$

*Moreover, with probability at least $1 - \delta$,*

$$\big|\mathbb{E}_{z_{\mathrm{mc}}}[\mathcal{H}_1] - \mathcal{H}_1\big| \leq \frac{\sqrt{288}\, C_G^2 R^2}{\eta_{t_l}^2 \sigma_{t_l}^2}\sqrt{\frac{\log(4/\delta)}{M}} + \frac{24d C_G^2}{\eta_{t_l}^2 \sigma_{t_l}^2 M}.$$

*Proof.* Using $x_{t_{l-1}}^{(i)} = \mu_{t_l}(x_{t_l}, s_\theta) + \sigma_{t_l} z_l^{(i)}$, we have for each $i$

$$H_i = h(x_0^{(i)}, 0)\, \frac{1}{\sigma_{t_l}} \nabla_{x_{t_l}} \mu_{t_l}(x_{t_l}, s_\theta)\, z_l^{(i)}.$$

Define $\Delta(x_{t_l}) = \nabla\widehat{h}(x_{t_l}, t_l) - \nabla h(x_{t_l}, t_l)$.

Let

$$F = \mathbb{E}_{X_{t_l} \sim P_{\theta, t_l}^h}\big[\|\Delta(X_t)\|_2^2\big], \qquad \mathcal{H}_1 = 2\eta_{t_l}^{-2} F.$$

Let $z \sim \mathcal{N}(0, I_d)$. The standard $\chi^2$ tail bound yields for any $s > 0$:

$$\mathbb{P}\Big(\|z\|_2^2 \geq d + 2\sqrt{ds} + 2s\Big) \leq e^{-s}.$$

With $s = \log \frac{2M}{\delta}$ and $R^2 = d + 2\sqrt{ds} + 2s$, a union bound gives

$$\mathbb{P}(\Omega_R^c) = \mathbb{P}\Big(\max_{i \leq M} \|z_l^{(i)}\|_2 > R\Big) \leq \sum_{i=1}^{M} \mathbb{P}(\|z_l^{(i)}\|_2 > R) \leq Me^{-s} = \frac{\delta}{2}.$$

On $\Omega_R$, since $\|\nabla_{x_{t_l}} \mu_{t_l}(x_{t_l}, s_\theta)\|_2 \leq C_G$,

$$\|H_i\|_2 \leq \frac{1}{\sigma_{t_l}}\|\nabla_{x_{t_l}} \mu_{t_l}(x_{t_l}, s_\theta)\|_2\|z_l^{(i)}\|_2 \leq \frac{C_G R}{\sigma_{t_l}}, \qquad \forall i.$$

Define $B = \frac{C_G R}{\sigma_{t_l}}$, consequently,

$$\Big\|\frac{1}{M}\sum_{j=1}^{M} H_j\Big\|_2 \leq B, \qquad \|\nabla h(x_{t_l}, t_l)\| \leq \frac{C_G}{\sigma_{t_l}} \leq \frac{C_G}{\sigma_{t_l}}R = B, \qquad \Rightarrow \qquad \|\Delta(x_{t_l})\|_2 \leq 2B,$$

all on $\Omega_R$.

Now on $\Omega_R$, replace only the $i$-th MC randomness block $\{Z_j^{(i)}\}_{j=1}^l$ by an independent copy, yielding $H_i'$, $F'$ and

$$\Delta'(x_{t_l}) = \Delta(x_{t_l}) + \delta(x_{t_l}), \qquad \delta(x_{t_l}) = \frac{1}{M}\big(H_i' - H_i\big).$$

On $\Omega_R$ (for both the original and the replaced samples),

$$\|\delta(x_{t_l})\|_2 \leq \frac{1}{M}\big(\|H_i'\|_2 + \|H_i\|_2\big) \leq \frac{2B}{M}.$$

Using $\big|\|a+b\|_2^2 - \|a\|_2^2\big| \leq 2\|a\|_2\|b\|_2 + \|b\|_2^2$ and $\|\Delta(x_{t_l})\| \leq 2B$, we get

$$\big|\|\Delta'(x_{t_l})\|_2^2 - \|\Delta(x_{t_l})\|_2^2\big| \leq 2\|\Delta(x_{t_l})\|_2\|\delta(x_{t_l})\|_2 + \|\delta(x_{t_l})\|_2^2 \leq 2(2B)\frac{2B}{M} + \left(\frac{2B}{M}\right)^2 \leq \frac{12B^2}{M}.$$

Taking $\mathbb{E}_{X_t}$ preserves the bound, so on $\Omega_R$,

$$|F - F^{(i)}| \leq c_i, \qquad c_i = \frac{12B^2}{M}.$$

Conditional on $\Omega_R = \bigcap_{1 \leq i \leq M}\{\|z_l^{(i)}\|_2 \leq R\}$, $\{\{Z_i^{(m)}\}_{i=1}^l\}_{m=1}^M$ are still mutually independent, McDiarmid's inequality gives for any $\varepsilon > 0$,

$$\mathbb{P}\big(|F - \mathbb{E}_{z_{\mathrm{mc}}}[F|\Omega_R]| \geq \varepsilon \mid \Omega_R\big) \leq 2\exp\left(-\frac{2\varepsilon^2}{\sum_{i=1}^M c_i^2}\right).$$

Since $\sum_{i=1}^M c_i^2 = M \cdot (12B^2/M)^2 = 144B^4/M$, we obtain

$$\mathbb{P}\big(|F - \mathbb{E}_{z_{\mathrm{mc}}}[F|\Omega_R]| \geq \varepsilon \mid \Omega_R\big) \leq 2\exp\left(-\frac{M\varepsilon^2}{72B^4}\right).$$

Set $\delta_1 = \delta/2$ and choose

$$\varepsilon = B^2\sqrt{\frac{72\log(2/\delta_1)}{M}} = B^2\sqrt{\frac{72\log(4/\delta)}{M}},$$

so that the RHS is at most $\delta_1 = \delta/2$. Multiplying by $2\eta_{t_l}^{-2}$ yields,

$$\mathbb{P}(|\mathcal{H}_1 - \mathbb{E}_{z_{\mathrm{mc}}}[\mathcal{H}_1|\Omega_R]| \geq 2\eta_{t_l}^{-2}\varepsilon) \leq \mathbb{P}(|\mathcal{H}_1 - \mathbb{E}_{z_{\mathrm{mc}}}[\mathcal{H}_1|\Omega_R]| \geq 2\eta_{t_l}^{-2}\varepsilon|\Omega_R) + \mathbb{P}(\Omega_R^c)$$
$$\leq \frac{\delta}{2} + \frac{\delta}{2}$$
$$= \delta,$$

where

$$2\eta_{t_l}^{-2}\varepsilon = \frac{\sqrt{288}\,C_G^2 R^2}{\eta_{t_l}^2 \sigma_{t_l}^2}\sqrt{\frac{\log(4/\delta)}{M}}.$$

Since

$$\big|\mathbb{E}_{z_{\mathrm{mc}}}[\mathcal{H}_1|\Omega_R] - \mathbb{E}_{z_{\mathrm{mc}}}[\mathcal{H}_1]\big| \leq \mathbb{E}_{z_{\mathrm{mc}}}[\mathcal{H}_1] + \mathbb{E}_{z_{\mathrm{mc}}}[\mathcal{H}_1|\Omega_R] \leq \left(1 + \frac{1}{P(\Omega_R)}\right)\mathbb{E}_{z_{\mathrm{mc}}}[\mathcal{H}_1] \leq 3\mathbb{E}_{z_{\mathrm{mc}}}[\mathcal{H}_1]$$

On $\Omega_R$, by triangle inequality,

$$|\mathcal{H}_1 - \mathbb{E}_{z_{\mathrm{mc}}}[\mathcal{H}_1]| \leq |\mathcal{H}_1 - \mathbb{E}_{z_{\mathrm{mc}}}[\mathcal{H}_1|\Omega_R]| + |\mathbb{E}_{z_{\mathrm{mc}}}[\mathcal{H}_1|\Omega_R] - \mathbb{E}_{z_{\mathrm{mc}}}[\mathcal{H}_1]|.$$

Since $\mathbb{E}_{z_{\mathrm{mc}}}[\mathcal{H}_1] \leq \frac{dC_G^2}{\eta_{t_l}^2 \sigma_{t_l}^2 M}$, then we can conclude, the following inequality holds with probability at least $1 - \delta$

$$\big|\mathbb{E}_{z_{\mathrm{mc}}}[\mathcal{H}_1] - \mathcal{H}_1\big| \leq \big|\mathbb{E}_{z_{\mathrm{mc}}}[\mathcal{H}_1|\Omega_R] - \mathcal{H}_1\big| + \big|\mathbb{E}_{z_{\mathrm{mc}}}[\mathcal{H}_1|\Omega_R] - \mathbb{E}_{z_{\mathrm{mc}}}[\mathcal{H}_1]\big|$$
$$\leq \frac{\sqrt{288}\,C_G^2 R^2}{\eta_{t_l}^2 \sigma_{t_l}^2}\sqrt{\frac{\log(4/\delta)}{M}} + \frac{24dC_G^2}{\eta_{t_l}^2 \sigma_{t_l}^2 M}. \tag{30}$$

$\square$

**Bounding $|\mathcal{H}_2 - \mathbb{E}_{z_{\mathrm{mc}}}[\mathcal{H}_2]|$ with high probability.** Define the constant $C_{\mathcal{H}_2} = \frac{4C_G^2}{\sigma_{t_l}^2 \eta_{t_l}^2 \rho}$, Then

$$\mathcal{H}_2 = C_{\mathcal{H}_2} \, \mathbb{E}_{X_{t_l} \sim P_{\theta, t_l}} \left[ (h(X_{t_l}, t_l) - \widehat{h}(X_{t_l}, t_l))^2 \right].$$

**Lemma B.3** (McDiarmid concentration for $\mathcal{H}_2$). *For any $\delta \in (0, 1)$, with probability at least $1 - \delta$ over the MC randomness,*

$$\left| \mathcal{H}_2 - \mathbb{E}_{z_{\mathrm{mc}}}[\mathcal{H}_2] \right| \leq C_{\mathcal{H}_2} \frac{3}{\sqrt{2M}} \sqrt{\log \frac{2}{\delta}} = \frac{12 C_G^2}{\sigma_{t_l}^2 \eta_{t_l}^2 \rho} \sqrt{\frac{\log(2/\delta)}{2M}}.$$

*Proof.* Introduce

$$Y(x_{t_l}) = \widehat{h}(x_{t_l}, t_l) - h(x_{t_l}, t_l), \qquad \bar{F} = \mathbb{E}_{X_{t_l} \sim P_{\theta, t_l}} \left[ Y(X_{t_l})^2 \right],$$

so that $\mathcal{H}_2 = C_{\mathcal{H}_2} \bar{F}$.

Fix an index $i \in \{1, \ldots, M\}$ and consider replace only the $i$-th MC randomness block $\{Z_j^{(i)}\}_{j=1}^l$ by an independent copy, producing $\widehat{h}'(x_{t_l}, t_l)$, $Y'(x_{t_l})$ and $\bar{F}'$. Define

$$\Delta(x_{t_l}) = \widehat{h}'(x_{t_l}, t_l) - \widehat{h}(x_{t_l}, t_l) = \frac{h((x_0^{(i)})', 0) - h(x_0^{(i)}, 0)}{M}.$$

Since $h((x_0^{(i)})', 0), h(x_0^{(i)}, 0) \in [0, 1]$, we have for all $x_{t_l}$,

$$|\Delta(x_{t_l})| \leq \frac{1}{M}.$$

Moreover, since $\widehat{h}(x_{t_l}, t_l) \in [0, 1]$ and $h(x_{t_l}, t_l) \in [0, 1]$, it follows that

$$Y(x_{t_l}) = \widehat{h}(x_{t_l}, t_l) - h(x_{t_l}, t_l) \in [-1, 1], \qquad \Rightarrow \qquad |Y(x_{t_l})| \leq 1 \quad \text{for all } x_{t_l}.$$

Let $Y'(x_{t_l}) = \widehat{h}'(x_{t_l}, t_l) - h(x_{t_l}, t_l) = Y(x_{t_l}) + \Delta(x_{t_l})$. Then we have

$$Y'(x_{t_l})^2 - Y(x_{t_l})^2 = \left( Y(x_{t_l}) + \Delta(x_{t_l}) \right)^2 - Y(x_{t_l})^2 = 2Y(x_{t_l}) \Delta(x_{t_l}) + \Delta(x_{t_l})^2.$$

Therefore,

$$\left| Y'(x_{t_l})^2 - Y(x_{t_l})^2 \right| \leq 2|Y(x_{t_l})| \, |\Delta(x_{t_l})| + |\Delta(x_{t_l})|^2 \leq 2 \cdot 1 \cdot \frac{1}{M} + \frac{1}{M^2} \leq \frac{3}{M}.$$

Taking expectation over $X_{t_l} \sim P_{\theta, t_l}$ preserves the bound, hence

$$|\bar{F}' - \bar{F}| = \left| \mathbb{E}_{X_{t_l} \sim P_{\theta, t_l}} \left[ Y'(X_{t_l})^2 - Y(X_{t_l})^2 \right] \right| \leq \mathbb{E}_{X_{t_l} \sim P_{\theta, t_l}} \left| Y'(X_{t_l})^2 - Y(X_{t_l})^2 \right| \leq \frac{3}{M}.$$

Thus $\bar{F}$ satisfies bounded differences with constants $\bar{c}_i = 3/M$ for all $i$. By McDiarmid's inequality, for any $\varepsilon > 0$,

$$\mathbb{P}\left( |\bar{F} - \mathbb{E}_{z_{\mathrm{mc}}}[\bar{F}]| \geq \varepsilon \right) \leq 2 \exp\left( -\frac{2\varepsilon^2}{\sum_{i=1}^M \bar{c}_i^2} \right).$$

Since $\sum_{i=1}^M \bar{c}_i^2 = M \cdot (3/M)^2 = 9/M$, we obtain

$$\mathbb{P}\left( |\bar{F} - \mathbb{E}_{z_{\mathrm{mc}}}[\bar{F}]| \geq \varepsilon \right) \leq 2 \exp\left( -\frac{2M}{9} \varepsilon^2 \right).$$

Because $\mathcal{H}_2 = C_{\mathcal{H}_2} \bar{F}$,

$$\mathbb{P}\left( |\mathcal{H}_2 - \mathbb{E}[\mathcal{H}_2]| \geq \tau \right) = \mathbb{P}\left( |\bar{F} - \mathbb{E}[\bar{F}]| \geq \tau/C_{\mathcal{H}_2} \right) \leq 2 \exp\left( -\frac{2M}{9} \left( \frac{\tau}{C_{\mathcal{H}_2}} \right)^2 \right),$$

which proves the tail bound. Setting the right-hand side equal to $\delta$ gives

$$|\mathcal{H}_2 - \mathbb{E}[\mathcal{H}_2]| \leq C_{\mathcal{H}_2} \sqrt{\frac{9}{2M} \log \frac{2}{\delta}} = \frac{12 C_G^2}{\sigma_{t_l}^2 \eta_{t_l}^2 \rho} \frac{3}{\sqrt{2M}} \sqrt{\log \frac{2}{\delta}}, \tag{31}$$

with probability at least $1 - \delta$, completing the proof. $\qquad \square$

**Bounding $\mathcal{H}_3$**  Set $\eta_{t_l} \leq 1/e$, by Lemma B.1 and Assumption 5.1, we have

$$
\begin{aligned}
\mathcal{H}_3 &\leq \int_{h(x_{t_l}, t_l) \leq \eta_{t_l}} \frac{8 \sup_{x_{t_l}} \|\nabla_{t_l} \mu_\theta(x_{t_l}, s_\theta)\|^2}{\sigma_{t_l}^2} \log \frac{1}{h(x_{t_l}, t_l)} \frac{p_{\theta, t_l}(x_{t_l}) h(x_{t_l}, t_l)}{\mathbb{P}(\mathcal{E}_{\bar{X}_0})} \mathrm{d}x_{t_l} \\
&\leq \int_{h(x_{t_l}, t_l) \leq \eta_{t_l}} \frac{8 C_G^2}{\sigma_{t_l}^2} \log \frac{1}{\eta_{t_l}} \frac{p_{\theta, t_l}(x_{t_l}) \eta_{t_l}}{\mathbb{P}(\mathcal{E}_{\bar{X}_0})} \mathrm{d}x_{t_l} \\
&\leq \frac{8 C_G^2 \eta_{t_l}}{\rho \sigma_{t_l}^2} \log \frac{1}{\eta_{t_l}}.
\end{aligned}
\tag{32}
$$

Putting all together,

$$
\begin{aligned}
&\mathbb{E}_{X_{t_l} \sim P_{\theta, t_l}^h} \left[ \|\nabla \log \widehat{h}(X_{t_l}, t_l) - \nabla \log h(X_{t_l}, t_l)\|_2^2 \right] \\
&\leq \mathcal{H}_1 + \mathcal{H}_2 + \mathcal{H}_3 \\
&\leq \mathcal{H}_3 + \mathbb{E}_{z_{\mathrm{mc}}}[\mathcal{H}_1 + \mathcal{H}_2] + |\mathcal{H}_1 - \mathbb{E}_{z_{\mathrm{mc}}}[\mathcal{H}_1]| + |\mathcal{H}_2 - \mathbb{E}_{z_{\mathrm{mc}}}[\mathcal{H}_2]| \\
&\leq \frac{8 C_G^2 \eta_{t_l}}{\rho \sigma_{t_l}^2} \log \frac{1}{\eta_{t_l}} + \frac{8 d C_G^2}{\eta_{t_l}^2 \sigma_{t_l}^2 M} + \frac{C_G^2}{M \sigma_{t_l}^2 \eta_{t_l}^2 \rho} + |\mathcal{H}_1 - \mathbb{E}_{z_{\mathrm{mc}}}[\mathcal{H}_1]| + |\mathcal{H}_2 - \mathbb{E}_{z_{\mathrm{mc}}}[\mathcal{H}_2]|.
\end{aligned}
$$

By high probability bounds for $|\mathcal{H}_1 - \mathbb{E}_{z_{\mathrm{mc}}}[\mathcal{H}_1]|$ and $|\mathcal{H}_2 - \mathbb{E}_{z_{\mathrm{mc}}}[\mathcal{H}_2]|$ in (30) and (31), the following inequality holds with probability at least $1 - 2\delta$,

$$
\begin{aligned}
&\mathbb{E}_{X_{t_l} \sim P_{\theta, t_l}^h} \left[ \|\nabla \log \widehat{h}(X_{t_l}, t_l) - \nabla \log h(X_{t_l}, t_l)\|_2^2 \right] \\
&\lesssim \frac{C_G^2 \eta_{t_l}}{\rho \sigma_{t_l}^2} \log \frac{1}{\eta_{t_l}} + \frac{d C_G^2}{\eta_{t_l}^2 \sigma_{t_l}^2 M} + \frac{C_G^2}{M \sigma_{t_l}^2 \eta_{t_l}^2 \rho} + \frac{C_G^2}{\sigma_{t_l}^2 \eta_{t_l}^2 \rho} \sqrt{\frac{\log(2/\delta)}{M}} \\
&\quad + \frac{C_G^2 R^2}{\eta_{t_l}^2 \sigma_{t_l}^2} \sqrt{\frac{\log(4/\delta)}{M}} \\
&\lesssim \frac{\eta_{t_l}}{\sigma_{t_l}^2} \log \frac{1}{\eta_{t_l}} + \frac{1}{\eta_{t_l}^2 \sigma_{t_l}^2} \left( \frac{1}{M} + \sqrt{\frac{\log(2/\delta)}{M}} + \log \frac{2M}{\delta} \sqrt{\frac{\log(4/\delta)}{M}} \right) \\
&\lesssim \frac{\eta_{t_l}}{\sigma_{t_l}^2} \log \frac{1}{\eta_{t_l}} + \frac{1}{\eta_{t_l}^2 \sigma_{t_l}^2} \left( \log \frac{2M}{\delta} \sqrt{\frac{\log(4/\delta)}{M}} \right).
\end{aligned}
$$

Here, $M$ is sufficiently large to satisfy $M \geq C_M$, where $C_M$ depends on $C_G$ and other relevant constants. Choose $\eta_{t_l} = \min\left(\frac{1}{M^{1/6}}, \frac{1}{e}\right) = M^{-1/6}$, then

$$
\begin{aligned}
&\mathbb{E}_{X_{t_l} \sim P_{\theta, t_l}^h} \left[ \|\nabla \log \widehat{h}(X_{t_l}, t_l) - \nabla \log h(X_{t_l}, t_l)\|_2^2 \right] \\
&\lesssim \frac{1}{\sigma_{t_l}^2 M^{1/6}} \log M + \frac{1}{M^{1/6} \sigma_{t_l}^2} \left( \log \frac{2M}{\delta} \sqrt{\log(4/\delta)} \right) \\
&\lesssim \frac{1}{M^{1/6} \sigma_{t_l}^2} \left( \log \frac{M}{\delta} \sqrt{\log(1/\delta)} \right)
\end{aligned}
$$

holds with probability at least $1 - 2\delta$. It also imply

$$
\begin{aligned}
&\mathbb{E}_{X_{t_l} \sim P_{\theta, t_l}^h} \left[ \|\nabla \log \widehat{h}(X_{t_l}, t_l) - \nabla \log h(X_{t_l}, t_l)\|_2^2 \right] \\
&= \mathcal{O}\left( \frac{1}{\sigma_{t_l}^2} \frac{\log \frac{M}{\delta} \sqrt{\log(1/\delta)}}{M^{1/6}} \right)
\end{aligned}
$$

holds with probability at least $1 - \delta$. $\qquad\square$

## B.4. Proof of Theorem 5.4

*Proof.* Since total variation satisfies the triangle inequality, we have

$$
\mathrm{TV}\left(P_{\mathrm{data}}(\cdot | \mathcal{E}_{X_0}), \widehat{P}_\theta^h\right) \leq \mathrm{TV}\left(P_{\mathrm{data}}(\cdot | \mathcal{E}_{X_0}), P_\theta(\cdot | \mathcal{E}_{\bar{X}_0})\right) + \mathrm{TV}\left(P_\theta(\cdot | \mathcal{E}_{\bar{X}_0}), \widehat{P}_\theta^h\right).
$$

The term $\mathrm{TV}\big(P_{\mathrm{data}}(\cdot|\mathcal{E}_{X_0}), P_\theta(\cdot|\mathcal{E}_{\bar{X}_0})\big)$ measures the modeling error between the ideal conditional data distribution and the conditional generated distribution, while $\mathrm{TV}\Big(P_\theta(\cdot|\mathcal{E}_{\bar{X}_0}), \widehat{P}_\theta^h\Big)$ measures the sampling error between the conditional generated distribution and the output distribution of `DOIT-Proto`.

**Bounding** $\mathrm{TV}\big(P_{\mathrm{data}}(\cdot|\mathcal{E}_{X_0}), P_\theta(\cdot|\mathcal{E}_{\bar{X}_0})\big)$   Define $h_{\mathrm{data}}$ as the corresponding $h$-function defined by $\mathcal{E}_{X_0}$. By definition, $\mathcal{E}_{X_0}$ are defined analogously by replacing $\bar{X}_0$ with $X_0$, then $h_{\mathrm{data}}(x,0) = h(x,0)$.

$$
\begin{aligned}
\mathrm{TV}\big(P_{\mathrm{data}}(\cdot|\mathcal{E}_{X_0}), P_\theta(\cdot|\mathcal{E}_{\bar{X}_0})\big) &= \frac{1}{2}\int_x \left| \frac{h(x,0)p_{\theta,0}(x)}{\mathbb{P}(\mathcal{E}_{\bar{X}_0})} - \frac{p_{\mathrm{data}}(x)h_{\mathrm{data}}(x,0)}{\mathbb{P}(\mathcal{E}_{X_0})} \right| \mathrm{d}x \\
&= \frac{1}{2\,\mathbb{P}(\mathcal{E}_{\bar{X}_0})\mathbb{P}(\mathcal{E}_{X_0})} \int_x h(x,0)\left|p_{\theta,0}(x)\mathbb{P}(\mathcal{E}_{X_0}) - p_{\mathrm{data}}(x)\mathbb{P}(\mathcal{E}_{\bar{X}_0})\right| \mathrm{d}x \\
&\leq \frac{1}{2\,\mathbb{P}(\mathcal{E}_{\bar{X}_0})} \int_x |p_{\theta,0}(x) - p_{\mathrm{data}}(x)| \, \mathrm{d}x \\
&\quad + \frac{1}{2\,\mathbb{P}(\mathcal{E}_{\bar{X}_0})\mathbb{P}(\mathcal{E}_{X_0})} \int \left|\mathbb{P}(\mathcal{E}_{\bar{X}_0}) - \mathbb{P}(\mathcal{E}_{X_0})\right| h(x,0)p_{\mathrm{data}}(x)\mathrm{d}x \\
&\lesssim \frac{1}{\rho}\mathrm{TV}(P_{\mathrm{data}}, P_\theta).
\end{aligned}
$$

Here $p_{\mathrm{data}}(x)$ is the density of $P_{\mathrm{data}}$.

**Bounding** $\mathrm{TV}\Big(P_\theta(\cdot|\mathcal{E}_{\bar{X}_0}), \widehat{P}_\theta^h\Big)$   By definition, $P_\theta(\cdot|\mathcal{E}_{\bar{X}_0})$ is the terminal distribution of (7),

$$
\mathrm{d}\bar{X}_t^h = \left[-\tfrac{1}{2}\bar{X}_t^h - s_\theta(\bar{X}_{t_l}^h, t_l) - \nabla \log h(\bar{X}_t^h, t)\right]\mathrm{d}t + \mathrm{d}\overline{W}_t, \tag{33}
$$

and $\bar{X}_0^h \sim P_\theta(\cdot|\mathcal{E}_{\bar{X}_0})$.

Meanwhile $\widehat{P}_\theta^h$ is the terminal distribution of the following SDE,

$$
\mathrm{d}\bar{X}_t^h = \left[-\tfrac{1}{2}\bar{X}_t^h - s_\theta(\bar{X}_{t_l}^h, t_l) - \nabla \log \widehat{h}(\bar{X}_{t_l}^h, t_l)\right]\mathrm{d}t + \mathrm{d}\overline{W}_t, \tag{34}
$$

where $\bar{X}_0^h \sim \widehat{P}_\theta^h$.

We use Theorem 9 in (Chen et al., 2023b) to bound $\mathrm{TV}\Big(P_\theta(\cdot|\mathcal{E}_{\bar{X}_0}), \widehat{P}_\theta^h\Big)$. Firstly we check

$$
\begin{aligned}
&\sum_{l=1}^L \mathbb{E}_{\mathbb{P}_\theta^h} \int_{t_{l-1}}^{t_l} \left[\|\nabla \log \widehat{h}(\bar{X}_{t_l}^h, t_l) - \nabla \log h(\bar{X}_t^h, t)\|_2^2\right] \mathrm{d}t \\
&\leq 2\sum_{l=1}^L \mathbb{E}_{\mathbb{P}_\theta^h} \int_{t_{l-1}}^{t_l} \left[\|\nabla \log h(\bar{X}_{t_l}^h, t_l) - \nabla \log h(\bar{X}_t^h, t)\|_2^2\right] \mathrm{d}t \\
&\quad + 2\sum_{l=1}^L \mathbb{E}_{\mathbb{P}_\theta^h} \int_{t_{l-1}}^{t_l} \left[\|\nabla \log h(\bar{X}_{t_l}^h, t_l) - \nabla \log \widehat{h}(\bar{X}_{t_l}^h, t_l)\|_2^2\right] \mathrm{d}t \\
&\lesssim \varepsilon_{\mathrm{dis}}T + \frac{\log\frac{LM}{\delta}\sqrt{\log(L/\delta)}}{M^{1/6}} \frac{T}{L}\sum_{l=1}^L \sigma_{t_l}^{-2} \\
&< \infty
\end{aligned}
$$

holds with probability at $1 - \frac{\delta L}{L} = 1 - \delta$ by Lemma 5.2 (For each discretization index $l \in \{1, \ldots, L\}$, the MC approximation bound holds with probability at least $1 - \delta/L$).

Then using the same argument in Theorem 9 in (Chen et al., 2023b), we can conclude

$$
\begin{aligned}
\mathrm{KL}(P_\theta(\cdot|\mathcal{E}_{\bar{X}_0}), \widehat{P}_\theta^h) &\lesssim \varepsilon_{\mathrm{dis}}T + \frac{\log\frac{LM}{\delta}\sqrt{\log(L/\delta)}}{M^{1/6}} \frac{T}{L}\sum_{l=1}^L \sigma_{t_l}^{-2} \\
&\lesssim \varepsilon_{\mathrm{dis}}T + \frac{\kappa_\sigma \log\frac{M}{\delta}\sqrt{\log(1/\delta)}}{M^{1/6}},
\end{aligned}
$$

where $\frac{T}{L}\sum_{l=1}^{L} 1/\sigma_{t_l}^2$ denotes as $\kappa_\sigma$. By Pinsker's Inequality,

$$\mathrm{TV}(P_\theta(\cdot|\mathcal{E}_{\bar{X}_0}), \widehat{P}_\theta^h) \lesssim \sqrt{\mathrm{KL}(P_\theta(\cdot|\mathcal{E}_{\bar{X}_0}), \widehat{P}_\theta^h)}$$

$$\lesssim \sqrt{\varepsilon_{\mathrm{dis}}T + \frac{\kappa_\sigma \log\frac{M}{\delta}\sqrt{\log(1/\delta)}}{M^{1/6}}}$$

holds with probability at $1 - \delta$.

Then we can conclude

$$\mathrm{TV}\left(P_{\mathrm{data}}(\cdot|\mathcal{E}_{X_0}), \widehat{P}_\theta^h\right) \leq \mathrm{TV}\left(P_{\mathrm{data}}(\cdot|\mathcal{E}_{X_0}), P_\theta(\cdot|\mathcal{E}_{\bar{X}_0})\right) + \mathrm{TV}\left(P_\theta(\cdot|\mathcal{E}_{\bar{X}_0}), \widehat{P}_\theta^h\right)$$

$$\lesssim \frac{1}{\rho}\mathrm{TV}(P_{\mathrm{data}}, P_\theta) + \sqrt{\varepsilon_{\mathrm{dis}}T + \frac{\kappa_\sigma \log\frac{M}{\delta}\sqrt{\log(1/\delta)}}{M^{1/6}}}.$$

$\square$

## B.5. Proof of Lemma B.1

*Proof.* By definition, $h(x_{t_l}, t_l) = \int h(x_{t_{l-1}}, t_{l-1})\phi_\theta(x_{t_{l-1}}|x_{t_l})\mathrm{d}x_{t_{l-1}}$.

We define the reweighed distribution $Q_\theta(X_{t_{l-1}}|X_{t_l})$ with density $q_\theta(x_{t_{l-1}}|x_{t_l}) = \frac{h(x_{t_{l-1}}, t_{l-1})\phi_\theta(x_{t_{l-1}}|x_{t_l})}{h(x_{t_l}, t_l)}$, then we have

$$\nabla_{x_{t_l}} \log h(x_{t_l}, t_l) = \int \frac{1}{\sigma_{t_l}^2}\nabla_{x_{t_l}}(\mu_{t_l}(x_{t_l}, s_\theta))(x_{t_{l-1}} - \mu_{t_l}(x_{t_l}, s_\theta))q_\theta(x_{t_{l-1}}|x_{t_l})\mathrm{d}x_{t_{l-1}}$$

$$= \frac{1}{\sigma_{t_l}^2}\nabla_{x_{t_l}}(\mu_{t_l}(x_{t_l}, s_\theta))\left[\mathbb{E}_{X_{t_{l-1}}\sim Q_\theta(\cdot|x_{t_l})}[X_{t_{l-1}}] - \mu_{t_l}(x_{t_l}, s_\theta)\right]$$

$$= \frac{1}{\sigma_{t_l}^2}\nabla_{x_{t_l}}(\mu_{t_l}(x_{t_l}, s_\theta))\left[\mathbb{E}_{X_{t_{l-1}}\sim Q_\theta(\cdot|x_{t_l})}[X_{t_{l-1}}] - \mathbb{E}_{X_{t_{l-1}}\sim \Phi(\cdot|x_{t_l})}[X_{t_{l-1}}]\right], \tag{35}$$

where $\Phi(\cdot|x_t)$ is $\mathcal{N}(\mu_{t_l}(x_{t_l}, s_\theta), \sigma_{t_l}^2 I)$.

It's obvious that $q_\theta \ll \phi_\theta$. By Talagrand's transportation inequality (Talagrand, 1996; Djellout et al., 2004; Xie et al., 2025), we have

$$\|\mathbb{E}_{X_{t_{l-1}}\sim Q_\theta(\cdot|x_{t_l})}[X_{t_{l-1}}] - \mathbb{E}_{X_{t_{l-1}}\sim \Phi(\cdot|x_{t_l})}[X_{t_{l-1}}]\|_2^2 \leq W_2^2(Q_\theta(\cdot|x_{t_l}), \Phi_\theta(\cdot|x_{t_l})) \leq 2\sigma_{t_l}^2 \mathrm{KL}(Q_\theta(\cdot|x_{t_l}), \Phi_\theta(\cdot|x_{t_l})).$$
$$\tag{36}$$

Now we turn to compute the KL divergence between $Q_\theta(\cdot|x_{t_l})$ and $\Phi_\theta(\cdot|x_{t_l})$

$$\mathrm{KL}(Q_\theta(\cdot|x_{t_l})||\Phi_\theta(\cdot|x_{t_l})) = \int q_\theta(x_{t_{l-1}}|x_{t_l}) \log\frac{q_\theta(x_{t_{l-1}}|x_{t_l})}{\phi_\theta(x_{t_{l-1}}|x_{t_l})}\mathrm{d}x_{t_{l-1}}$$

$$= \int q_\theta(x_{t_{l-1}}|x_{t_l}) \log\frac{h(x_{t_{l-1}}, t_{l-1})}{h(x_{t_l}, t_l)}\mathrm{d}x_{t_{l-1}}$$

$$= \int q_\theta(x_{t_{l-1}}|x_{t_l})(\log h(x_{t_{l-1}}, t_{l-1}) - \log h(x_{t_l}, t_l))\mathrm{d}x_{t_{l-1}}$$

$$\leq -\log h(x_{t_l}, t_l). \tag{37}$$

By combining (35), (36), (37), we can conclude

$$\|\nabla_{x_{t_l}} \log h(x_{t_l}, t_l)\|_2 \leq \frac{\|\nabla_{x_{t_l}}(\mu_{t_l}(x_{t_l}, s_\theta))\|_2}{\sigma_{t_l}}\sqrt{2\log\frac{1}{h(x_{t_l}, t_l)}}.$$

$\square$

# C. Experimental Details and Additional Results

## C.1. Formal Construction of the Conditioning Event $\mathcal{E}_{\bar{X}_0}$

In Section 6.1, we defined the terminal $h$-function as $h(x, 0) \propto \exp(r(x)/\tau)$. Here, we rigorously define the event $\mathcal{E}_{\bar{X}_0}$ that induces this function by invoking Lemma 3.2.

The target distribution is defined via exponential tilting as $q(x) \propto p_{\theta,0}(x) \exp(r(x)/\tau)$. To satisfy the boundedness condition in Lemma 3.2, we identify the upper bound of the unnormalized density ratio:

$$\frac{q(x)}{p_{\theta,0}(x)} \propto \exp\left(\frac{r(x)}{\tau}\right) \leq \exp\left(\frac{r_{\max}}{\tau}\right) = C_q,$$

where $r_{\max} \geq \sup_x r(x)$. Following Lemma 3.2, we introduce an auxiliary variable $U \sim \mathrm{Unif}(0,1)$ and define the event:

$$\mathcal{E}_{\bar{X}_0} = \left\{ U \leq \frac{\exp(r(\bar{X}_0)/\tau)}{C_q} \right\} = \left\{ U \leq \exp\left(\frac{r(\bar{X}_0) - r_{\max}}{\tau}\right) \right\}.$$

The resulting $h$-function is then:

$$h(x,0) = \mathbb{P}(\mathcal{E}_{\bar{X}_0} \mid \bar{X}_0 = x) = \exp\left(\frac{r(x) - r_{\max}}{\tau}\right) \propto \exp\left(\frac{r(x)}{\tau}\right).$$

This confirms that our choice of $h$-function corresponds to a valid conditioning event under the bounded reward assumption.

## C.2. Experiments on Improving Aesthetic Scores

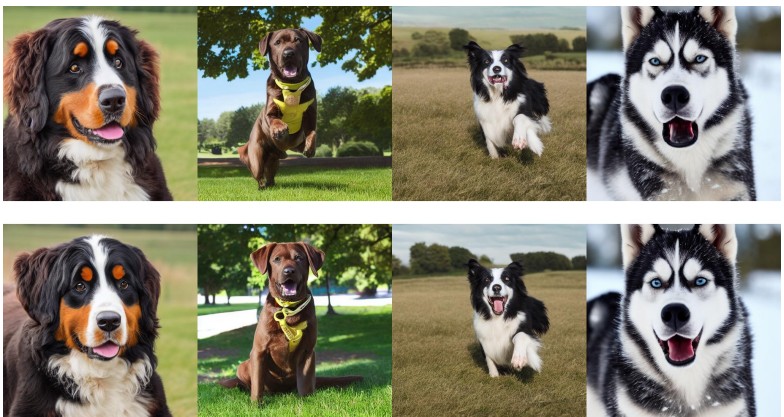

*Figure 3.* Comparison of dog images generated by `Stable Diffusion v1.5`. The upper row displays samples from vanilla generation process, while the bottom row shows samples guided by `DOIT-Practical` using the aesthetic score as the reward.

### C.2.1. EXPERIMENTAL SETUP

For the generation process, we utilize $L = 20$ diffusion steps, employing the Euler ancestral sampler (Karras et al., 2022). To balance correction effectiveness with computational cost, Doob correction is applied only up to a cutoff timestep $l^* = 10$. The approximation of $\nabla \log h$ is performed using $M = 32$ Monte Carlo samples per step.

### C.2.2. ABLATION STUDY ON $\tau$ AND $\gamma$

We conduct an ablation study to analyze the impact of the temperature parameter $\tau$ and the strength $\gamma$. Tables 4 through 8 detail the statistical summaries of the generated aesthetic scores (minimum, quantiles, mean, and maximum) across various $(\tau, \gamma)$ configurations.

Our analysis reveals a trade-off between optimization intensity and stability. Specifically, the combination of a low temperature (small $\tau$) and high correction strength (large $\gamma$) results in aggressive optimization. However, overly aggressive settings (e.g., $\tau = 0.2, \gamma = 12$) can destabilize the mechanism, leading to potential collapse. Based on these empirical results, we identify a robust operating regime with $\tau \in [0.3, 0.4]$ and $\gamma \in [4.0, 8.0]$.

It is important to note that these optimal hyperparameters are specific to the current configuration. Altering the base model or sampler (e.g., using DDIM with stochasticity parameter $\eta > 0$) or the reward feedback may shift the stability region, necessitating further hyperparameter tuning.

## C.3. Experiments on Improving ImageReward with Reweighting and Resampling

Following Zhang et al. (2025b), we build upon the official codebases of FK-Steering (Singhal et al., 2025) and DAS (Kim et al., 2025) to implement our baselines and sampling routines. We evaluate performance using the standard ImageReward

*Table 4.* Minimum aesthetic score.

| Doob $\gamma \setminus \tau$ | 0.2 | 0.3 | 0.4 | 0.5 |
|---|---|---|---|---|
| 0.0 | 5.688 ± 0.142 | 5.688 ± 0.142 | 5.688 ± 0.142 | 5.688 ± 0.142 |
| 1.0 | 5.728 ± 0.242 | 5.763 ± 0.116 | 5.733 ± 0.128 | 5.736 ± 0.137 |
| 2.0 | 5.646 ± 0.417 | 5.740 ± 0.238 | 5.752 ± 0.152 | 5.741 ± 0.140 |
| 3.0 | 5.782 ± 0.414 | 5.856 ± 0.227 | 5.760 ± 0.245 | 5.791 ± 0.167 |
| 4.0 | 5.665 ± 0.345 | 5.757 ± 0.256 | 5.804 ± 0.218 | 5.794 ± 0.185 |
| 5.0 | 5.004 ± 0.316 | 5.821 ± 0.226 | 5.877 ± 0.195 | 5.836 ± 0.263 |
| 6.0 | 4.432 ± 0.159 | 5.921 ± 0.137 | 5.893 ± 0.176 | 5.847 ± 0.247 |
| 7.0 | 3.856 ± 0.469 | 5.661 ± 0.416 | 5.870 ± 0.170 | 5.907 ± 0.188 |
| 8.0 | 3.717 ± 0.239 | 5.480 ± 0.495 | 5.868 ± 0.209 | 5.840 ± 0.134 |
| 9.0 | 3.094 ± 0.607 | 5.392 ± 0.177 | 5.971 ± 0.096 | 5.901 ± 0.159 |
| 10.0 | 3.188 ± 0.216 | 4.389 ± 0.148 | 5.837 ± 0.052 | 5.638 ± 0.324 |
| 11.0 | 3.296 ± 0.238 | 4.264 ± 0.465 | 5.763 ± 0.045 | 5.967 ± 0.073 |
| 12.0 | 2.967 ± 0.137 | 4.157 ± 0.239 | 5.250 ± 0.351 | 5.948 ± 0.106 |

*Table 5.* First quartile (Q1) of aesthetic score.

| Doob $\gamma \setminus \tau$ | 0.2 | 0.3 | 0.4 | 0.5 |
|---|---|---|---|---|
| 0.0 | 6.157 ± 0.041 | 6.157 ± 0.041 | 6.157 ± 0.041 | 6.157 ± 0.041 |
| 1.0 | 6.200 ± 0.050 | 6.199 ± 0.038 | 6.193 ± 0.038 | 6.188 ± 0.034 |
| 2.0 | 6.270 ± 0.042 | 6.216 ± 0.040 | 6.209 ± 0.031 | 6.208 ± 0.026 |
| 3.0 | 6.368 ± 0.019 | 6.282 ± 0.022 | 6.244 ± 0.032 | 6.235 ± 0.037 |
| 4.0 | 6.433 ± 0.028 | 6.323 ± 0.017 | 6.272 ± 0.024 | 6.265 ± 0.031 |
| 5.0 | 6.252 ± 0.101 | 6.393 ± 0.008 | 6.311 ± 0.011 | 6.282 ± 0.015 |
| 6.0 | 5.816 ± 0.250 | 6.452 ± 0.028 | 6.361 ± 0.024 | 6.310 ± 0.012 |
| 7.0 | 5.400 ± 0.119 | 6.453 ± 0.032 | 6.371 ± 0.012 | 6.319 ± 0.030 |
| 8.0 | 4.920 ± 0.117 | 6.422 ± 0.036 | 6.452 ± 0.043 | 6.358 ± 0.011 |
| 9.0 | 4.691 ± 0.104 | 6.210 ± 0.142 | 6.451 ± 0.029 | 6.388 ± 0.028 |
| 10.0 | 4.385 ± 0.088 | 5.790 ± 0.223 | 6.484 ± 0.032 | 6.433 ± 0.049 |
| 11.0 | 4.300 ± 0.067 | 5.477 ± 0.103 | 6.408 ± 0.040 | 6.471 ± 0.035 |
| 12.0 | 3.989 ± 0.078 | 5.275 ± 0.144 | 6.255 ± 0.038 | 6.463 ± 0.032 |

*Table 6.* Mean aesthetic score.

| Doob $\gamma \setminus \tau$ | 0.2 | 0.3 | 0.4 | 0.5 |
|---|---|---|---|---|
| 0.0 | 6.357 ± 0.029 | 6.357 ± 0.029 | 6.357 ± 0.029 | 6.357 ± 0.029 |
| 1.0 | 6.452 ± 0.025 | 6.421 ± 0.025 | 6.403 ± 0.024 | 6.396 ± 0.026 |
| 2.0 | 6.525 ± 0.033 | 6.472 ± 0.023 | 6.453 ± 0.024 | 6.434 ± 0.024 |
| 3.0 | 6.600 ± 0.037 | 6.537 ± 0.028 | 6.490 ± 0.018 | 6.470 ± 0.023 |
| 4.0 | 6.645 ± 0.022 | 6.576 ± 0.033 | 6.520 ± 0.028 | 6.498 ± 0.018 |
| 5.0 | 6.476 ± 0.097 | 6.635 ± 0.036 | 6.569 ± 0.035 | 6.527 ± 0.023 |
| 6.0 | 6.098 ± 0.123 | 6.691 ± 0.041 | 6.611 ± 0.042 | 6.559 ± 0.037 |
| 7.0 | 5.736 ± 0.057 | 6.696 ± 0.077 | 6.652 ± 0.039 | 6.584 ± 0.035 |
| 8.0 | 5.351 ± 0.099 | 6.631 ± 0.057 | 6.691 ± 0.050 | 6.625 ± 0.036 |
| 9.0 | 5.042 ± 0.074 | 6.393 ± 0.089 | 6.702 ± 0.036 | 6.643 ± 0.042 |
| 10.0 | 4.823 ± 0.082 | 6.054 ± 0.079 | 6.695 ± 0.040 | 6.671 ± 0.041 |
| 11.0 | 4.698 ± 0.019 | 5.853 ± 0.062 | 6.596 ± 0.047 | 6.707 ± 0.036 |
| 12.0 | 4.440 ± 0.066 | 5.674 ± 0.067 | 6.442 ± 0.049 | 6.701 ± 0.036 |

*Table 7.* Third quartile (Q3) of aesthetic score.

| Doob $\gamma \setminus \tau$ | 0.2 | 0.3 | 0.4 | 0.5 |
|---|---|---|---|---|
| 0.0 | 6.560 ± 0.037 | 6.560 ± 0.037 | 6.560 ± 0.037 | 6.560 ± 0.037 |
| 1.0 | 6.674 ± 0.027 | 6.635 ± 0.031 | 6.617 ± 0.030 | 6.596 ± 0.047 |
| 2.0 | 6.787 ± 0.013 | 6.676 ± 0.023 | 6.655 ± 0.048 | 6.639 ± 0.051 |
| 3.0 | 6.798 ± 0.080 | 6.736 ± 0.039 | 6.691 ± 0.031 | 6.679 ± 0.057 |
| 4.0 | 6.852 ± 0.065 | 6.846 ± 0.070 | 6.728 ± 0.041 | 6.691 ± 0.035 |
| 5.0 | 6.810 ± 0.054 | 6.857 ± 0.028 | 6.831 ± 0.054 | 6.719 ± 0.038 |
| 6.0 | 6.536 ± 0.017 | 6.951 ± 0.039 | 6.852 ± 0.062 | 6.774 ± 0.034 |
| 7.0 | 6.208 ± 0.084 | 6.897 ± 0.071 | 6.908 ± 0.013 | 6.816 ± 0.054 |
| 8.0 | 5.860 ± 0.094 | 6.866 ± 0.062 | 6.925 ± 0.083 | 6.871 ± 0.059 |
| 9.0 | 5.514 ± 0.090 | 6.655 ± 0.078 | 6.980 ± 0.100 | 6.884 ± 0.057 |
| 10.0 | 5.290 ± 0.102 | 6.432 ± 0.070 | 6.950 ± 0.046 | 6.918 ± 0.101 |
| 11.0 | 5.097 ± 0.050 | 6.286 ± 0.069 | 6.838 ± 0.086 | 6.909 ± 0.084 |
| 12.0 | 4.907 ± 0.060 | 6.172 ± 0.060 | 6.732 ± 0.064 | 6.899 ± 0.027 |

*Table 8.* Maximum aesthetic score.

| Doob $\gamma \setminus \tau$ | 0.2 | 0.3 | 0.4 | 0.5 |
|---|---|---|---|---|
| 0.0 | 7.028 ± 0.091 | 7.028 ± 0.091 | 7.028 ± 0.091 | 7.028 ± 0.091 |
| 1.0 | 7.161 ± 0.139 | 7.125 ± 0.148 | 7.085 ± 0.111 | 7.078 ± 0.115 |
| 2.0 | 7.291 ± 0.219 | 7.197 ± 0.185 | 7.164 ± 0.161 | 7.111 ± 0.165 |
| 3.0 | 7.363 ± 0.192 | 7.296 ± 0.147 | 7.185 ± 0.169 | 7.183 ± 0.179 |
| 4.0 | 7.292 ± 0.073 | 7.338 ± 0.187 | 7.251 ± 0.112 | 7.241 ± 0.156 |
| 5.0 | 7.299 ± 0.125 | 7.385 ± 0.170 | 7.323 ± 0.133 | 7.288 ± 0.162 |
| 6.0 | 6.966 ± 0.122 | 7.326 ± 0.156 | 7.320 ± 0.185 | 7.331 ± 0.210 |
| 7.0 | 6.649 ± 0.154 | 7.407 ± 0.222 | 7.371 ± 0.161 | 7.302 ± 0.169 |
| 8.0 | 6.699 ± 0.054 | 7.310 ± 0.094 | 7.498 ± 0.301 | 7.333 ± 0.161 |
| 9.0 | 6.446 ± 0.065 | 7.131 ± 0.185 | 7.463 ± 0.292 | 7.299 ± 0.153 |
| 10.0 | 6.210 ± 0.547 | 6.936 ± 0.093 | 7.273 ± 0.147 | 7.387 ± 0.064 |
| 11.0 | 5.835 ± 0.069 | 6.850 ± 0.151 | 7.336 ± 0.081 | 7.314 ± 0.131 |
| 12.0 | 5.516 ± 0.075 | 6.852 ± 0.025 | 7.179 ± 0.181 | 7.317 ± 0.175 |

prompt set (Singhal et al., 2025), utilizing $L = 100$ total sampling steps and candidate set sizes of $K \in \{4, 8\}$. The sampler used in this experiment is chosen as DDIM (Song et al., 2020a) with the stochasticity parameter $\eta = 1.0$. All results are reported as the mean and standard deviation over 4 independent trials.

For baseline configurations, we strictly adhere to the settings reported in Zhang et al. (2025b), which replicate the original authors' implementations; we do not perform additional tuning for these baselines. Specifically, the BFS resampling schedule is defined by the interval $\mathcal{L} = [20, 80]$ with a frequency of $l_{\text{freq}} = 20$. We configure the BFS parameters as $\Omega_{\text{BFS}} = (10, \texttt{Increase}, \texttt{Max}, \texttt{SSP})$, corresponding to the search temperature, scoring method, buffer update strategy, and resampling algorithm, respectively. Regarding `DOIT-Practical`-specific hyperparameters, we employ $M = 32$ Monte Carlo samples, a cutoff time $l^* = 20$, a temperature $\tau = 0.2$, and a correction strength $\gamma = 0.8$. Algorithm 3 details the complete integration of our proposed method with the BFS framework.

### C.4. Offline RL Experiments

**Setup.** We adopt the experimental framework of Lu et al. (2023), utilizing their pre-trained diffusion policy and ground-truth Q-functions (as reward oracles). The diffusion model is conditioned on the state $s$ to generate actions $a$, employing a $L = 15$ steps DDIM sampler.

We evaluate our method on the D4RL locomotion benchmark (Fu et al., 2020), which spans three MuJoCo environments and three dataset compositions. Each environment represents a standard continuous-control task, while the datasets vary in the quality and diversity of the constituent trajectories.

**Environments and Datasets.** The specific tasks are defined as follows:

- **HalfCheetah:** A planar running task that rewards maximizing forward velocity.

- **Hopper:** A one-legged hopping task requiring balance and forward progress.

**Algorithm 3** `DOIT-Practical` + BFS Integration

**Input:** Pre-trained diffusion model $s_\theta(x, t)$; Doob correction strength $\gamma$; Monte Carlo samples $M$; Time threshold $l^* \in [L]$; Number of parallel trajectories $K$; Resampling interval $\mathcal{L}$; Resampling frequency $l_{\text{freq}}$; BFS parameters $\Omega_{\text{BFS}}$.

1: Initialize resampling scores $\{b_{\text{prev}}^{(k)}\}_{k=1}^K$.
2: Sample $\{x_{t_L}^{(k)}\}_{k=1}^K$ from $\mathcal{N}(0, I)$.
3: **for** $l = L, L-1, \ldots, 1$ **do**
4:   **if** $l \in \mathcal{L}$ and $(l \pmod{l_{\text{freq}}} == 0)$ **then**
5:     $\{x_{t_l}^{(k)}\}_{k=1}^K, \{b_{\text{prev}}^{(k)}\}_{k=1}^K \leftarrow \text{BFS}(\{x_{t_l}^{(k)}\}_{k=1}^K; \Omega_{\text{BFS}})$
6:   **end if**
7:   **for** $k = 1, \ldots, K$ **do**
8:     **if** $1 < l \le l^*$ **then**
9:       Sample $\{x_{t_{l-1}}^{(m)}\}_{m=1}^M$ from $\mathcal{N}(\mu_{t_l}(x_{t_l}, s_\theta), \sigma_{t_l}^2 I)$.
10:       Compute $\{\widehat{x}_0^{(m)}\}_{m=1}^M$ via (10).
11:       Compute $\nabla \log \widehat{h}(x_{t_l}^{(k)}, t_l)$ in (9) via $\{\widehat{x}_0^{(m)}\}_{m=1}^M$.
12:       $\nabla \log \widehat{p}_\theta^h(x_{t_l}^{(k)}) \leftarrow s_\theta(x_{t_l}^{(k)}, t_l) + \gamma \nabla \log \widehat{h}(x_{t_l}^{(k)}, t_l)$.
13:       Sample $x_{t_{l-1}}$ from $\mathcal{N}(\mu(x_{t_l}, \nabla \log \widehat{p}_\theta^h), \sigma_{t_l}^2 I)$.
14:     **else**
15:       Sample $x_{t_{l-1}}$ from $\mathcal{N}(\mu(x_{t_l}, s_\theta), \sigma_{t_l}^2 I)$.
16:     **end if**
17:   **end for**
18: **end for**
19: **Return** $x_0 = \arg\max_{k \in \{1, \ldots, K\}} r(x_0^{(k)})$.

- **Walker2d:** A bipedal walking task rewarding stable forward locomotion.

The datasets are categorized by the nature of the policy used for data collection:

- **Medium-Expert:** A mixture of expert-level and medium-level policies' decision data.
- **Medium:** Decision data generated by a single medium-level policy.
- **Medium-Replay:** Diverse decision data generated by a large set of medium-level policies.

*Table 9.* Hyperparameter settings for our method across datasets and environments. $\eta$ denotes the DDIM stochasticity parameter, $\gamma$ the Doob correction scale, $\tau$ the temperature, $t^\star$ the correction start timestep, and $K$ the number of candidate samples selected via best-of-$K$.

| Dataset | Environment | $\eta$ | $\gamma$ | $\tau$ | $t^\star$ | $K$ |
|---|---|---|---|---|---|---|
| Medium-Expert | HalfCheetah | 0.7 | 0.25 | 0.5 | 8 | 4 |
| Medium-Expert | Hopper | 0.8 | 0.25 | 0.7 | 4 | 4 |
| Medium-Expert | Walker2d | 0.4 | 0.5 | 0.4 | 10 | 8 |
| Medium | HalfCheetah | 0.2 | 0.25 | 0.5 | 10 | 4 |
| Medium | Hopper | 0.6 | 0.75 | 0.4 | 8 | 8 |
| Medium | Walker2d | 0.3 | 0.5 | 0.3 | 10 | 4 |
| Medium-Replay | HalfCheetah | 0.4 | 0.5 | 0.3 | 10 | 8 |
| Medium-Replay | Hopper | 0.2 | 0.5 | 0.5 | 6 | 4 |
| Medium-Replay | Walker2d | 0.2 | 0.25 | 0.7 | 8 | 8 |

**Hyperparameter Settings.** To ensure a fair comparison, we align our evaluation protocol with Zhang et al. (2025b), using their codebase and hyperparameter search strategy for baseline methods. A key metric for fairness is the computational budget, particularly for methods involving test-time search (sampling $K$ candidates and selecting the best-of-$K$).

The baselines employ various resource-intensive strategies: TFG (Ye et al., 2024) allows up to 8 recurrence steps; SVDD (Li et al., 2024) employs up to $K = 16$ particles and $M = 32$ Monte Carlo candidates, selecting the optimal temperature $\alpha$ from the set $\{0.0, 0.1, \ldots, 0.6\}$; DAS (Kim et al., 2025) uses $K = 16$ particles; and TTS (Zhang et al., 2025b) uses up to $K = 4$ particles combined with optimized correction strength and iterative & recurrence steps. To maintain a comparable computational budget, we configure our method with $M = 32$ lookahead samples (for gradient estimation) and generate up to $K = 8$ final candidates. We note that unlike the baselines, our method does not require iterative recurrence or optimization steps during sampling.

We perform a hyperparameter search over the following grids: correction strength $\gamma \in \{0.25, 0.5, 0.75, 1\}$, temperature $\tau \in \{0.3, 0.4, 0.5, 0.6, 0.7\}$, and cutoff time $l^* \in \{4, 6, 8, 10\}$. Additionally, since `DOIT-Practical` and SVDD rely on the variance of the transition kernel (which vanishes in deterministic sampling), we tune the DDIM stochasticity parameter $\eta \in \{0.2, 0.3, 0.4, 0.5, 0.6, 0.7, 0.8\}$. For fair comparison we evaluate our method on different seeds used for hyperparameter search. The optimal hyperparameters of `DOIT-Practical` for each task are reported in Table 9.

## C.5. An Extension of the Convergence Guarantee to the Surrogate Approximation

In Section 5, our convergence guarantee is established for `DOIT-Proto`, where the Doob correction is approximated by full backward Monte Carlo rollouts. In practice, `DOIT-Practical` replaces the terminal samples obtained from full rollouts by the one-step surrogate in (10), which avoids additional score-network evaluations. This subsection records a simple conditional extension showing how the analysis changes if the additional surrogate error is controlled.

Let $\widehat{h}$ denote the full-rollout Monte Carlo estimator used in `DOIT-Proto`, and let $\widetilde{h}$ denote the surrogate estimator used in `DOIT-Practical`. Throughout this subsection, $\nabla \log \widetilde{h}$ denotes the surrogate correction vector produced by `DOIT-Practical` through the same plug-in ratio form as (9), with the terminal rollout samples replaced by their one-step surrogates. Let $\widetilde{P}_\theta^h$ denote the terminal distribution of the sampling process driven by this surrogate correction,

$$d\widetilde{X}_t^h = \left[ -\frac{1}{2}\widetilde{X}_t^h - s_\theta(\widetilde{X}_{t_l}^h, t_l) - \nabla \log \widetilde{h}(\widetilde{X}_{t_l}^h, t_l) \right] dt + dW_t, \qquad t \in [t_{l-1}, t_l]. \tag{38}$$

The following assumption quantifies only the extra error introduced by replacing full rollouts with the surrogate construction.

**Assumption C.1** (Surrogate approximation error). *There exists $\varepsilon_{\mathrm{surr}} \geq 0$ such that, for every $l \in \{1, \ldots, L\}$,*

$$\mathbb{E}_{X_{t_l} \sim P_{\theta, t_l}^h} \left[ \left\| \nabla \log \widetilde{h}(X_{t_l}, t_l) - \nabla \log \widehat{h}(X_{t_l}, t_l) \right\|_2^2 \right] \leq \varepsilon_{\mathrm{surr}}. \tag{39}$$

Assumption C.1 should be interpreted as a stability condition on the one-step surrogate used by `DOIT-Practical`. It separates the surrogate error from the Monte Carlo approximation error analyzed in Lemma 5.2. When $\varepsilon_{\mathrm{surr}} = 0$, the bound below reduces to the same form as Theorem 5.4 up to constants.

**Corollary C.2** (Conditional guarantee under a surrogate-error assumption). *Suppose Assumptions 5.1, 5.3, and C.1 hold, and choose the truncation level $\eta_t$ as in Lemma 5.2. Then, with probability at least $1 - \delta$, it holds that*

$$\mathrm{TV}\left( P_{\mathrm{data}}(\cdot | \mathcal{E}_{X_0}), \widetilde{P}_\theta^h \right) \lesssim \frac{1}{\rho} \mathrm{TV}(P_{\mathrm{data}}, P_\theta) + \sqrt{\varepsilon_{\mathrm{dis}} T + \varepsilon_{\mathrm{surr}} T + \frac{\kappa_\sigma \log(M/\delta)\sqrt{\log(1/\delta)}}{M^{1/6}}}, \tag{40}$$

*where $\kappa_\sigma = \frac{T}{L} \sum_{l=1}^{L} \sigma_{t_l}^{-2}$.*

For clarity, this corollary analyzes only the error caused by replacing full terminal rollouts with one-step surrogates. It does not cover the additional implementation heuristics in Algorithm 2, such as the correction strength $\gamma$ and the time threshold $l^*$.

*Proof.* The proof follows the same structure as the proof of Theorem 5.4. We first decompose the total variation distance into a modeling error term and a sampling error term. By the triangle inequality,

$$\mathrm{TV}\left( P_{\mathrm{data}}(\cdot | \mathcal{E}_{X_0}), \widetilde{P}_\theta^h \right)$$
$$\leq \mathrm{TV}\left( P_{\mathrm{data}}(\cdot | \mathcal{E}_{X_0}), P_\theta(\cdot | \mathcal{E}_{\bar{X}_0}) \right) + \mathrm{TV}\left( P_\theta(\cdot | \mathcal{E}_{\bar{X}_0}), \widetilde{P}_\theta^h \right). \tag{41}$$

The first term is the same modeling error as in Theorem 5.4. By the same argument used there, since $\mathbb{P}(\mathcal{E}_{\bar{X}_0}) \geq \rho$, we have

$$\mathrm{TV}\left(P_{\mathrm{data}}(\cdot|\mathcal{E}_{X_0}), P_\theta(\cdot|\mathcal{E}_{\bar{X}_0})\right) \lesssim \frac{1}{\rho}\mathrm{TV}(P_{\mathrm{data}}, P_\theta). \tag{42}$$

It remains to control the sampling error $\mathrm{TV}(P_\theta(\cdot|\mathcal{E}_{\bar{X}_0}), \widetilde{P}_\theta^h)$. By definition, $P_\theta(\cdot|\mathcal{E}_{\bar{X}_0})$ is the terminal distribution of (7),

$$\mathrm{d}\bar{X}_t^h = \left[-\tfrac{1}{2}\bar{X}_t^h - s_\theta(\bar{X}_{t_l}^h, t_l) - \nabla \log h(\bar{X}_t^h, t)\right]\mathrm{d}t + \mathrm{d}\overline{W}_t, \tag{43}$$

and $\bar{X}_0^h \sim P_\theta(\cdot|\mathcal{E}_{\bar{X}_0})$. We use Theorem 9 in (Chen et al., 2023b) to bound $\mathrm{TV}\left(P_\theta(\cdot|\mathcal{E}_{\bar{X}_0}), \widetilde{P}_\theta^h\right)$. Firstly we check

$$\sum_{l=1}^L \mathbb{E}_{\mathbb{P}_\theta^h} \int_{t_{l-1}}^{t_l} \left[\|\nabla \log \widetilde{h}(\bar{X}_{t_l}^h, t_l) - \nabla \log h(\bar{X}_t^h, t)\|_2^2\right]\mathrm{d}t$$

$$\leq 2\sum_{l=1}^L \mathbb{E}_{\mathbb{P}_\theta^h} \int_{t_{l-1}}^{t_l} \left[\|\nabla \log h(\bar{X}_{t_l}^h, t_l) - \nabla \log h(\bar{X}_t^h, t)\|_2^2\right]\mathrm{d}t$$

$$+ 2\sum_{l=1}^L \mathbb{E}_{\mathbb{P}_\theta^h} \int_{t_{l-1}}^{t_l} \left[\|\nabla \log h(\bar{X}_{t_l}^h, t_l) - \nabla \log \widetilde{h}(\bar{X}_{t_l}^h, t_l)\|_2^2\right]\mathrm{d}t$$

$$\leq 2\sum_{l=1}^L \mathbb{E}_{\mathbb{P}_\theta^h} \int_{t_{l-1}}^{t_l} \left[\|\nabla \log h(\bar{X}_{t_l}^h, t_l) - \nabla \log h(\bar{X}_t^h, t)\|_2^2\right]\mathrm{d}t$$

$$+ 4\sum_{l=1}^L \mathbb{E}_{\mathbb{P}_\theta^h} \int_{t_{l-1}}^{t_l} \left[\|\nabla \log h(\bar{X}_{t_l}^h, t_l) - \nabla \log \widehat{h}(\bar{X}_{t_l}^h, t_l)\|_2^2\right]\mathrm{d}t$$

$$+ 4\sum_{l=1}^L \mathbb{E}_{\mathbb{P}_\theta^h} \int_{t_{l-1}}^{t_l} \left[\|\nabla \log \widehat{h}(\bar{X}_{t_l}^h, t_l) - \nabla \log \widetilde{h}(\bar{X}_{t_l}^h, t_l)\|_2^2\right]\mathrm{d}t$$

$$\lesssim \varepsilon_{\mathrm{surr}}T + \varepsilon_{\mathrm{dis}}T + \frac{\log \frac{LM}{\delta}\sqrt{\log(L/\delta)}}{M^{1/6}}\frac{T}{L}\sum_{l=1}^L \sigma_{t_l}^{-2}$$

$$< \infty$$

holds with probability at $1 - \frac{\delta L}{L} = 1 - \delta$ by Lemma 5.2 (For each discretization index $l \in \{1, \ldots, L\}$, the MC approximation bound holds with probability at least $1 - \delta/L$).

Then using the same argument in Theorem 9 in (Chen et al., 2023b), we can conclude

$$\mathrm{KL}(P_\theta(\cdot|\mathcal{E}_{\bar{X}_0}), \widetilde{P}_\theta^h) \lesssim \varepsilon_{\mathrm{surr}}T + \varepsilon_{\mathrm{dis}}T + \frac{\log \frac{LM}{\delta}\sqrt{\log(L/\delta)}}{M^{1/6}}\frac{T}{L}\sum_{l=1}^L \sigma_{t_l}^{-2}$$

$$\lesssim \varepsilon_{\mathrm{surr}}T + \varepsilon_{\mathrm{dis}}T + \frac{\kappa_\sigma \log \frac{M}{\delta}\sqrt{\log(1/\delta)}}{M^{1/6}},$$

where $\frac{T}{L}\sum_{l=1}^L 1/\sigma_{t_l}^2$ denotes as $\kappa_\sigma$. By Pinsker's Inequality,

$$\mathrm{TV}(P_\theta(\cdot|\mathcal{E}_{\bar{X}_0}), \widetilde{P}_\theta^h) \lesssim \sqrt{\mathrm{KL}(P_\theta(\cdot|\mathcal{E}_{\bar{X}_0}), \widetilde{P}_\theta^h)}$$

$$\lesssim \sqrt{\varepsilon_{\mathrm{dis}}T + \varepsilon_{\mathrm{surr}}T + \frac{\kappa_\sigma \log \frac{M}{\delta}\sqrt{\log(1/\delta)}}{M^{1/6}}}$$

holds with probability at $1 - \delta$.

Then we can conclude

$$\mathrm{TV}\left(P_{\mathrm{data}}(\cdot|\mathcal{E}_{X_0}), \widetilde{P}_\theta^h\right) \leq \mathrm{TV}\left(P_{\mathrm{data}}(\cdot|\mathcal{E}_{X_0}), P_\theta(\cdot|\mathcal{E}_{\bar{X}_0})\right) + \mathrm{TV}\left(P_\theta(\cdot|\mathcal{E}_{\bar{X}_0}), \widetilde{P}_\theta^h\right)$$

$$\lesssim \frac{1}{\rho}\mathrm{TV}(P_{\mathrm{data}}, P_\theta) + \sqrt{\varepsilon_{\mathrm{dis}}T + \varepsilon_{\mathrm{surr}}T + \frac{\kappa_\sigma \log \frac{M}{\delta}\sqrt{\log(1/\delta)}}{M^{1/6}}}.$$

$\square$

