# OpenReview forum: "Training-Free Adaptation of Diffusion Models via Doob's $h$-Transform"
_ICML.cc/2026/Conference — ICML 2026 regular_

### Official Review · Reviewer_z2aw · 2026-03-09

**Soundness:** 2
**Presentation:** 2
**Significance:** 3
**Originality:** 3
**Overall Recommendation:** 2
**Confidence:** 4

**Summary:**

This paper presents a new inference time method for reward tilting of pre-trained diffusion models. Via Doob’s $h$-transform, reward tilting of pre-trained diffusion models is the same as adding a correction term of form $\nabla \log h(t, X_t)$ to the time-reversed SDE. The paper suggests using a Monte-Carlo estimation to approximate the correction term at each sampling step. This algorithm is studied to show convergence bounds of the approximate correction to the true correction term as well as the distance between the true tilted distribution and the distribution obtained through the proposed algorithm. However, it is noted that this proposed algorithm is computationally expensive since $M$ SDE samples must be generated at each sampling step. Therefore, a “practical version” of the algorithm is introduced which is then used in the experiments.

**Compliance With Llm Reviewing Policy:**

Affirmed.

**Final Justification:**

My concerns about the presentation/soundness still remain. I agree revising the abstract and introduction and separating the two methods to make it clear that the analysis and convergence results don't hold for the surrogate would improve the paper. However, I find this to be a significant change requiring a new review, therefore I maintain my reject score.

**Key Questions For Authors:**

How does replacing samples of $X_0$ with $E[X_0 \mid X_{t_{l-1}}]$ affect the bounds in Theorem 5.4 and Lemma 5.2?

**Limitations:**

Yes.

**Strengths And Weaknesses:**

**Strengths:**

1. Presentation: I found it easy to follow and understand the presented method, and therefore I think the paper was clearly-written. Moreover, I think the work was correctly positioned in related literature.
2. Soundness: I believe the submission is technically sound in that the paper includes theoretical results with correct proofs. Furthermore, all assumptions are clearly stated and discussed.
3. Originality: The paper proposes a new algorithm for reward tilting by proposing an MC approximation to Doob’s $h$-transform.


**Weaknesses:**

1. I believe a critical weakness of the paper is the mismatch between the method that is actually applied in the experiments and the method introduced in the main paper for which the convergence results hold. That is, within the paper two algorithms are introduced. The first is computationally infeasible (as discussed in the paper) since at each step of the SDE sample, $M$ trajectories of the reverse SDE must be generated, so if there are $L$ sample steps, this leads to $M\cdot L!$ function evaluations. The second algorithm introduced replaces samples of $X_0$ with Tweedie approximations $E[X_0 \mid X_{t_{l-1}}]$. This ensures that the second method is computationally feasible. However, the paper is mainly formulated around Method 1 including the convergence results, which show the approximation error of the score and the distance between the learned distribution and true distribution. Method 2 is not introduced until the experiment section and is what ends up being used in comparisons. However the convergence and error results no longer hold since this is a different method. This is not clear from the paper narrative and I therefore believe the paper needs to be significantly restructured. For example, I believe it would make more sense to derive error results for Algorithm 2 which is what is used in the experiments.

2. Related to the previous point, it is not clear to me that Method 2, used in the experiments, will converge to the correct distribution. This does not have to be a problem, since in some situations one just wants to have higher reward samples and having the exact distribution is not so important. However, the paper claims a convergence guarantee which I find somewhat misleading.

3. Another weakness of the paper is the numerical stability of the method. The method involves dividing by the term $h(x, t)$ which can be near zero. The paper claims that the stability issue is due to the pre-trained model and not a limitation of the proposed model. However if the event or set $\mathcal{E}$ we are conditioning on is rare, then $h(x, t) = P(X_0 \in \mathcal{E}| X_t = x)$ will be small even assuming the pre-trained model is perfect, therefore I think it is a limitation of the proposed model. That said, the paper does discuss the potential division by zero and introduces a truncation level to avoid this and I appreciate this discussion being included in the paper.


The following issues are minor and do not affect my score.
- I believe the colours used for links are not the ones used in the ICML template.
- Line 077 should read: “for *a* non-differentiable reward” or “for non-differentiable reward*s*”
- Line 097: “*a* simulation-based…”
- 099: “*the* DOIT algorithm…”
- I believe there is a mistake in Algorithm 1. I think the trajectories should be simulated via (2) instead of (8). This would also be in line with Lemma 4.2.
- I do not understand the second part of the sentence: “Moreover, we establish a theoretical convergence guarantee of our method, which is highly limited with only very recent advances”.

**Summary**

In summary, I find the paper as written to be somewhat misleading since the paper provides analysis for a method that is different from the method used in the experiments. For this reason, I give lower scores for soundness and presentation. This is my main critique and due to this I recommend the paper be rejected.

---

> ### Author Rebuttal · Authors · 2026-03-31
>
> We sincerely appreciate the reviewer’s comments and suggestions. We respond below to the main weaknesses, questions, and limitations.
> > **W1 & Q1: A critical weakness of the paper is the mismatch between the method that is actually applied in the experiments and the method introduced in the main paper for which the convergence results hold. ...**
>
> **AW1 & AQ1:**
> The surrogate approximation is introduced to make the algorithm computationally feasible. That said, we respectfully disagree that this constitutes a mismatch in the sense of an incoherent paper narrative. Rather, this reflects the intended structure of our work: we first formulate the exact Doob $h$-transform correction and study a principled prototype (Alg. 1), then introduce Alg. 2 as a practical surrogate approximation of the same correction term to obtain an efficient implementation. In this sense, the paper follows a natural progression from principled formulation, to analyzable prototype, tocomputationally feasible approximation, rather than a disconnect between theory and experiments.
>
> The current theory in Thm. 5.4 is established for Alg. 1, while experiments use the surrogate implementation (Alg. 2) for computational efficiency. Alg. 2 replaces the terminal samples $x_0^{(m)}$ with the surrogate $\\hat x_0^{(m)}$, which introduces an additional approximation error not covered by Thm. 5.4 as currently stated. Our theoretical analysis extends to the surrogate approximation with very minor modification as summarized in the following corollary. In particular, we additionally assume
>
> **Assumption 5.5.**
> There exists $\\varepsilon\_\{\\mathrm{surr}\}\geq 0$, such that
> $$
> \\mathbb{E}_\{X\_\{t_l\}\\sim P^h\_\{\theta,t_l\}\} \\left[\\|\\nabla  \\log \\tilde{ h}\(X\_\{t_l\},t_l\) -\\nabla \\log \\hat h(X\_\{t_l\},t_l)\\|_2^2 \\right]\\leq\\varepsilon\_\{\\mathrm{surr}\}
> $$
> holds for any $l\\in \\{1,...L\\}$, where $\\hat h$ is the MC estimator in Alg. 1 and $\\tilde h$ is the surrogate estimator in Alg. 2. This quantifies the extra error introduced by the surrogate step.
>
> **Corollary 5.6.**  Suppose Assumptions 5.1, 5.3 and 5.5 hold and choose $\\eta_t$ as in Lemma 5.2.
> Then, with probability at least $1-\\delta$,
> $$
> \mathrm{TV}\\!\left(P_{\mathrm{data}}(\cdot \mid \mathcal E_{X_0}), \hat P_\theta^h\right)
> \lesssim
> \frac{1}{\rho}\mathrm{TV}(P_{\mathrm{data}},P_\theta)
> +
> \sqrt{
> \frac{\kappa_\sigma \log \frac{M}{\delta}\sqrt{\log(1/\delta)}}{M^{1/6}}
> +
> \varepsilon_{\mathrm{dis}}T
> +
> \varepsilon_{\mathrm{surr}}T
> },
> $$
> where $\\kappa_\sigma=\frac{T}{L}\sum_{l=1}^L \sigma_{t_l}^{-2}$.
>
> The proof follows the same template as Thm. 5.4, with a direct triangle-inequality argument introducing only one additional term, $\\varepsilon_{\\mathrm{surr}}T$, which scales linearly with $T$ and is relatively mild. Thus, while Thm. 5.4 does not directly cover Alg. 2 as stated, Alg. 2 can be incorporated via Corollary 5.6.
> > **W2: Related to the previous point, it is not clear that Method 2, used in experiments, will converge to the correct distribution.  ...**
>
> **AW2:** As explained in AW1, Alg. 2 can also be covered via **Cor. 5.6**.
>
> We therefore respectfully disagree this reflects a misleading. Our goal is to provide a self-contained and thorough study: we first analyze the exact Doob $h$-transform correction through a principled prototype (Alg. 1), then introduce Alg. 2 as a computationally efficient surrogate of the same correction term. In this sense, the paper follows a natural progression from principled formulation, to analyzable prototype, to efficient approximation.
> > **W3: Another weakness of the paper is the numerical stability of the method. ...**
>
> **AW3:** There might be some misunderstanding of limitation of the proposed method and the fundamental difficulty of the task. When the event is rare, it is a fundamentally difficult task for inference-time adaptation. This fundamental difficulty is well understood in offline reinforcement learning through the lens of exploration coverage ([1],[2]). In practice, we also use antithetic sampling to reduce MC variance and improve stability, although it does not remove the underlying rare-event difficulty.
> > **W4-8: Colours used for links are not the ones used in the ICML template. Typos in line 077,097, 099, and Alg. 1.**
>
> **AW4-8:** Thank you for pointing this out. We will fix them in the revised version.
> >**W9: I do not understand the second part of the sentence: “Moreover, we establish a theoretical convergence guarantee of our method, which is highly limited with only very recent advances”.**
>
> **AW9:** The second part of the sentence was intended to mean theoretical guarantees for Doob’s $h$-transform based training-free adaptation methods, especially convergence guarantees of the kind we establish, are still scarce and only appears in very recent work.
>
> [1] Chen and Jiang, “Information-theoretic considerations in batch reinforcement learning.”
>
> [2] Jin et al., “Is pessimism provably efficient for offline RL?”

---

> > ### Author Rebuttal · Reviewer_z2aw · 2026-04-01
> >
> > I thank the authors for their response and clarifications. However, my concerns about the surrogate implementation still remain.
> > If we assume that the difference between the MC estimator and surrogate model $\varepsilon_{\text{surr}}$ is small, then indeed it makes sense to effectively use the convergence results for the MC estimator as in Corollary 5.6 (with the additional error term accounting for $\varepsilon_{\text{surr}}$). However, my concern is rather that $\varepsilon_{\text{surr}}$ will be large, and I do not see why $\varepsilon_{\text{surr}}$ would converge. Therefore, both my concerns about the method converging to the correct distribution and more generally the error induced by Algorithm 2 remain.
> >
> > I think the surrogate method could still be useful without the convergence or error guarantees, however the paper is currently written as though these hold for the surrogate method, which I do not believe to be true. This could either be fixed by including an analysis on the $\varepsilon_{\text{surr}}$ or a substantial rewrite. For this reason I maintain my reject score.

---

> > > ### Author Response · Authors · 2026-04-06
> > >
> > > We sincerely thank the reviewer again for the careful clarification and follow-up.
> > >
> > > We appreciate several points in the reviewer’s comment that are in fact aligned with our paper. First, we appreciate the reviewer’s recognition that the convergence analysis for the MC estimator is meaningful. Second, we agree that, if one additionally controls the surrogate approximation error, then it is natural to extend the MC-based result with an extra surrogate-error term. Third, we also appreciate the reviewer’s acknowledgement that the surrogate implementation can still be useful in practice even without a standalone convergence or error guarantee. This is exactly the role Algorithm 2 plays in our paper: it is introduced as a practical surrogate motivated by computational efficiency.
> > >
> > > We agree that the current manuscript does not separate the two versions of the DOIT algorithm sharply enough. Concretely, Theorem 5.4 is proved for Algorithm 1, whereas Algorithm 2 is the practical surrogate version used in the experiments for computational efficiency. We will revise the abstract and introduction to make this scope explicit throughout. Our intended structure is to present an exact Doob correction, then an analyzable prototypical algorithm, then an efficient surrogate implementation. We view this theory-to-practice progression as a strength of the paper.
> > >
> > > We would also like to bring to the reviewer's attention that the clarification above does not affect the soundness of the theorem proved for Algorithm 1. The role of Algorithm 2 is practical efficiency, and we will present it strictly in that way. Empirically, the surrogate is computationally much cheaper, while achieving comparable performance to even full-trajectory simulation. For example, in Table 3 the surrogate and full simulation have nearly identical mean reward statistics (6.726 vs. 6.714), while runtime drops from 39.584s to 1.712s per image. We will make clear that this empirical evidence supports the practicality of Algorithm 2. We hope this clarification resolves the concern about the scope of our theoretical analysis.

---

### Official Review · Reviewer_HRZL · 2026-03-13

**Soundness:** 3
**Presentation:** 3
**Significance:** 3
**Originality:** 3
**Overall Recommendation:** 5
**Confidence:** 4

**Summary:**

This paper proposes a training-free alignment method with non-differentiable rewards for diffusion models.  The authors adopt Doob's h-transform for alignment and propose an importance sampling method for calculating the correction term. They provide a convergence guarantee for the proposed method and verify the effectiveness on text-to-image generation tasks and the D4RL locomotion benchmark.

**Compliance With Llm Reviewing Policy:**

Affirmed.

**Final Justification:**

The strength of this paper lies in its soundness and clarity. My main concern for this paper lies in significance. During the rebuttal, the authors provide a comparison with resampling methods, which somehow improves the significance.

**Key Questions For Authors:**

Is the label of the red dotline in Figure 1 not complete?

**Strengths And Weaknesses:**

Strengths:

1. The theoretical guarantee for the proposed method is novel, i.e., Theorem 5.4.
2. The performance on the D4RL benchmark is good.
3. Good design for "Surrogate" to reduce computational cost. Otherwise, the computational time for rollout M trajectory for updating one diffusion step is computationally expensive.

Weaknesses:
1. As mentioned in the limitation section, MC approximation of the correction term might suffer from high variance, which can be left for future work.
2. It would be better if the authors could provide some simulations to prove that the proposed method is better than rejection sampling.

---

> ### Author Rebuttal · Authors · 2026-03-31
>
> We sincerely appreciate the reviewer’s comments and suggestions. We respond below to the main weaknesses, questions, and limitations.
>
> > **W1: As mentioned in the limitations section, the MC approximation of the correction term might suffer from high variance, which can be left for future work.**
>
> **AW1:** We agree that the MC approximation of the correction term can have high variance when the target high-reward region is extremely rare. However, this is not merely an implementation issue of our method, but a fundamental difficulty of training-free inference-time adaptation toward rare events: when the event probability under the pre-trained model is extremely small, only a tiny fraction of rollouts provide useful signal for estimating $\\nabla \\log h$.
>
> This limitation is also tied to the quality of the pre-trained model. If the pre-trained model already places reasonable mass near the desired high-reward region, the correction can be estimated more stably; otherwise, any training-free sampling-based method will face substantial difficulty.
>
> In practice, we partially mitigate this issue via antithetic sampling, which reduces Monte Carlo variance and improves stability empirically, though it does not remove the underlying rare-event difficulty.
>
> > **W2: It would be better if the authors could provide some simulations to show that the proposed method is better than rejection sampling.**
>
> **AW2**: Thank you for the suggestion. We agree that a rejection sampling (RS) baseline is important for comparison. Our goal is to sample from
> $$q(x) \propto p_\theta(x)\exp(r(x)/\tau),$$
> where $r(x)$ is the reward and $\tau$ is the temperature. Since rewards are not normalized to $[0,1]$, we introduce a hyperparameter $r_{\max}$, and implement rejection sampling as follows:
>
> - While the number of accepted samples is less than $K$:
>   - Sample a full trajectory $x \sim p_\theta(x)$;
>   - Sample $U \sim \mathrm{Unif}(0,1)$;
>   - Accept if $U \le \min\\{1, \exp((r(x) - r_{\max})/\tau)\\}$, otherwise reject.
>
> We include the $\min\\{1, \cdot\\}$ safeguard to ensure the acceptance probability remains valid even when $r_{\max}$ may not strictly upper-bound all possible rewards.
>
> We evaluate this baseline under the same setting as Section 6.2 using Stable Diffusion v1.5, aiming to obtain $K=32$ high-reward samples. For DOIT, we use $\tau=0.4$ and correction strength $\gamma=7.0$, and reuse our existing results (Fig. 2 and Appendix C.2.2) for both DOIT and vanilla sampling. For fair comparison, we apply the same temperature $\tau=0.4$ to the RS baseline. We vary $r_\{\max\} \in \\{7.0, 7.25, 7.5\\}$ based on the observed reward range in Fig. 2. Here, Q1 denotes the first quartile and Q3 denotes the third quartile.
>
> ---
> |Method|Min|Q1|Mean|Q3|Max| NFEs ($\downarrow$)|
> |--------|-----|----|------|----|-----|-----|
> |Vanilla Sampling|5.69|6.16|6.36|6.56|7.03|**640**|
> |DOIT ($\gamma=8.0$)|5.87|**6.45**|**6.69**|**6.93**|**7.50**|**640**|
> |RS ($r_{\max}=7.0$)| **5.94**|6.42|6.56|6.72|7.19|2880|
> |RS($r_{\max}=7.25$)|**5.94**|6.48|6.62|6.80|7.31|5760|
> |RS($r_{\max}=7.5$)|**5.94**|6.46| 6.63|6.80|7.31|8000|
> ---
> From the table, DOIT consistently achieves superior performance across reward statistics (Q1, mean, Q3, and max) over rejection sampling. Regarding efficiency, as discussed in Section 6.1 (line 349), our practical DOIT implementation incurs **zero additional NFEs** compared to vanilla diffusion generation, by reusing the network forward pass for one-step surrogate approximation (Equation 10, line 347). In contrast, RS requires more full trajectory generations to obtain $K$ accepted samples, leading to significantly higher NFEs. As $r_{\max}$ increases, the acceptance probability $\exp((r(x) - r_{\max})/\tau)$ decreases, resulting in a higher rejection ratio and further increased computational cost.
>
> Overall, although rejection sampling improves over vanilla sampling, it requires substantially higher computational cost and still underperforms DOIT. In contrast, DOIT achieves **significantly better reward performance at comparable cost to vanilla sampling**, making it both more effective and more efficient than rejection sampling.
>
>
> > **Q1: Is the label of the red dotted line in Figure 1 incomplete?**
>
> **AQ1:** Thank you for pointing this out. The label in Figure 1 is meant to be read **together across the two legend rows**. More specifically, the legend entry should be read as
> “**MC Simulation via the pre-trained diffusion model $P_\theta$**,”
> rather than as two separate labels. In other words, both the red and blue sampled trajectories in Figure 1 are Monte Carlo trajectories generated from the pre-trained diffusion model, and they are used to estimate $\nabla \log h(x_{t_l}, t_l)$. The purple trajectory then denotes the DOIT tilted sampling process after incorporating this estimated correction.
>
> We agree that the current layout may cause confusion, and we will revise the figure/legend in the final version to make this clearer.

---

> > ### Author Rebuttal · Reviewer_HRZL · 2026-04-03
> >
> > Thanks for the authors' responses and I keep my current score.

---

> > > ### Author Response · Authors · 2026-04-06
> > >
> > > We sincerely thank the reviewer for the thoughtful follow-up. We are glad that our responses were helpful in addressing the concerns, and we greatly appreciate the reviewer’s continued positive assessment of the paper.

---

### Official Review · Reviewer_f81d · 2026-03-13

**Soundness:** 3
**Presentation:** 4
**Significance:** 3
**Originality:** 2
**Overall Recommendation:** 4
**Confidence:** 4

**Summary:**

This paper proposes a training free algorithm for inference time alignment of diffusion models. The main idea is using Doob's h-transform, and the paper proposes algorithms to estimate the $h$ function and its gradient via Monte Carlo simulation. Theorectical guarantees of the algorithms are provided, with empirical justification on text-to-image diffusion models and diffusion policies.

**Compliance With Llm Reviewing Policy:**

Affirmed.

**Final Justification:**

The authors have mainly addressed my concerns in the number of NFEs required for algorithm implementation, so it looks to me now that the comparison to BofK baselines is fair.

**Key Questions For Authors:**

Q1: Can authors count how many actual NFEs are used actually in the sampling of DOIT, and then plot the performance of best of N with N that yields the same amount of NFEs?

**Limitations:**

Yes.

**Strengths And Weaknesses:**

**Strengths:**

1. The paper is generally well written. Notations are clear, presentations are nice, with ideas illustrated pretty cleanly. The theoretical analysis is also pretty thorough. I enjoyed reading this paper.
2. The paper provides a new inference time scaling approach applicable for non differential reward.

**Weakness:**

1. The idea of using h transform has been explored in earlier works, and the MC estimation here is less novel.
2. In actual inference, the proposed algorithm needs to search among different trajectories to estimate $h$, which makes the algorithm less parrallizable for usage.
3. There is a significant missing piece of fair comparison between proposed DOIT with the baseline inference time methods like best of K. For example in Figure 2, there lacks the report of baseline performance by just choosing top pictures yet using the same amount of NFEs. This comparison is more than important to showcase the actual performance gain over BofK baselines.
4. Minor: The literature review seems to be less comprehensive. More recent works in RL for diffusion/flow models like are not included: like Score as Action (https://arxiv.org/pdf/2502.01819), Flow GRPO (https://arxiv.org/abs/2505.05470) etc.

---

> ### Author Rebuttal · Authors · 2026-03-31
>
> We sincerely appreciate the reviewer’s comments and suggestions. We respond below to the main weaknesses, questions, and limitations.
>
> > **W1: The idea of using the h-transform has been explored in earlier works, and the MC estimation here is less novel.**
>
> **AW1:** As discussed in page 2-3, line 103-116, we agree previous work, like [1], also use Doob's $h$-transform as a theoretical framework for training-free diffusion adaptation. Our novelty claim is therefore not that we introduce Doob's $h$-transform itself, but that we develop a novel approximation strategy for $\nabla \log h$ with a broad applicability to non-differentiable rewards and supporting theory.
>
> Specifically, our method approximates $\\nabla h$ and $h$ separately using MC samples, and then forms $\\nabla \\log h$ via a plug-in ratio estimator. A key implication is our method does not require differentiability of the reward function, which substantially broadens the class of rewards it can handle. Moreover, the underlying MC approximation error is theoretically controlled in our analysis.
>
> By contrast, as discussed on our page 4-5, lines 213-227, [1] uses
> $$
> \\nabla_{x_{t_l}} \\log h(x_{t_l},t_l) \\approx \\nabla_{x_{t_l}} \\log h(\\mathbb{E}[\\bar X_0 \\mid \\bar X_{t_l}=x_{t_l}],0),
> $$
> which requires differentiability of the terminal reward, equivalently of $h(\\cdot,0)$.
>
> Therefore, we respectfully disagree our contribution is merely an MC estimation variant. The key distinction is our approximation changes what reward classes can be handled: it supports non-differentiable rewards and is based on MC approximation with a provable approximation guarantee.
> > **W2: In actual inference, the proposed algorithm needs to search among different trajectories to estimate $h$, which makes the algorithm less parallelizable in practice.**
>
> **AW2:**  Both Alg. 1 and Alg. 2 are naturally parallelizable. In both cases, $M$ trajectories used to estimate $\\nabla \\log h$ are independent conditional rollouts and can be sampled fully in parallel as a batch, with only a final aggregation step, since the MC estimation does not introduce a sequential dependency across trajectories.
>
> Moreover, the sampling process itself can also be run in batch across multiple samples. Alg. 2 is especially favorable, since it replaces explicit terminal rollouts with the surrogate approximation in Eq. (10), thereby preserving essentially the same batched execution pattern as standard diffusion inference.
> > **W3: There is a significant missing piece in the fair comparison between the proposed DOIT method and baseline inference-time methods like best-of-$K$. For example, in Figure 2, the paper does not report the baseline performance obtained by simply selecting the top images while using the same amount of NFEs. This comparison is very important to demonstrate the actual performance gain over best-of-$K$ baselines.**
>
> **AW3:** We respectfully disagree with this concern.
>
> As discussed in Sec. 6.1 (line 349), our practical DOIT implementation (Alg. 2) incurs zero additional NFEs compared to vanilla diffusion generation by reusing the network forward pass for the one-step surrogate approximation (Eq. 10, line 347). Therefore, the computational cost of DOIT generating $K$ samples is directly comparable to vanilla generation that also generates $K$ samples (best-of-$K$ simply selects the best from these $K$ samples). This is why we did not explicitly report NFEs in Figure 2, as they are identical across these methods. For example, generating $K=32$ samples with 20 diffusion steps requires 640 NFEs for both DOIT and vanilla generation.
> > **W4: The literature review seems less comprehensive. More recent works in RL for diffusion/flow models are not included, such as *Score as Action* and *Flow GRPO*.**
>
> **AW4:** We thank the reviewer for pointing this out. The two papers focus on training-based fine-tuning of diffusion models, while we focus on training-free adaptation. We will add these two to the training-based adaptation part of the related work in the revision.
> > **Q1: Can authors count how many actual NFEs are used in DOIT sampling, and then plot the performance of best-of-$K$ with $K$ chosen so that it uses the same total amount of NFEs?**
>
> **AQ1:** As discussed in AW3, our practical DOIT implementation incurs zero additional NFEs compared to vanilla diffusion generation by reusing the network forward pass. Therefore, all results reported using DOIT in Sections 6.2 and 6.3, including image generation and D4RL experiments, operate under the same NFE budget as best-of-$K$ baselines.
>
> Under the same NFE budget, DOIT consistently outperforms best-of-$K$ across tasks. For example, as reported in Table 2, DOIT surpasses best-of-$K$ in all 9/9 tasks, improving the average final performance metric from 76.9 to 87.6.
>
> [1] Nguyen et al. "h-Edit: Effective and Flexible Diffusion-Based Editing via Doob's h-Transform."

---

> > ### Author Rebuttal · Reviewer_f81d · 2026-04-02
> >
> > Thanks for the authors' responses and most of my concerns are addressed. I will raise the score.

---

> > > ### Author Response · Authors · 2026-04-06
> > >
> > > We sincerely thank the reviewer for the positive follow-up. We are glad that our responses addressed most of the concerns, and we greatly appreciate the reviewer’s willingness to raise the score.

---

### Official Review · Reviewer_G14x · 2026-03-13

**Soundness:** 3
**Presentation:** 3
**Significance:** 2
**Originality:** 2
**Overall Recommendation:** 4
**Confidence:** 3

**Summary:**

This paper presents a training-free and computationally efficient method to adapt a pre-trained diffusion model to a reward function. It proposes DOIT (Doob-Oriented Inference-time Transformation), which frames adaptation as a measure transport problem from the original generative distribution to a reward-conditioned target distribution. The method demonstrates its effectiveness on offline reinforcement learning (D4RL) benchmarks.

**Compliance With Llm Reviewing Policy:**

Affirmed.

**Final Justification:**

The rebuttal strengthens the paper and addresses my main questions, and I remain positive about the paper.

**Key Questions For Authors:**

- The theoretical results in Section 5 apply to Algorithm 1, while the experiments use Algorithm 2. What are the theoretical implications of the surrogate approximation in Equation (10)?

- Can the bound in Theorem 5.4 be extended to cover this practical variant?

- What is the actual overhead relative to vanilla diffusion sampling?

- Does DOIT remain efficient for larger diffusion models (e.g., SDXL)?

**Limitations:**

The authors have discussed the sensitivity of the method to the probability of high-reward regions, which can lead to large variance in approximation.

**Strengths And Weaknesses:**

**Strengths:**
- The method achieves state-of-the-art results on D4RL offline RL benchmarks among inference-time adaptation methods

- The approximation in Algorithm 2 is a practical contribution that makes the method feasible by avoiding additional score network evaluations during inference.

- The paper attempts to provide an end-to-end convergence guarantee for a training-free adaptation method, which is a valuable

**Weaknesses:**

My main concern is that the core idea using Doob's h-transform to steer diffusion models at inference time has been explored in prior work. Specifically, Nguyen et al. (2025) also propose a training-free method utilizing the same theoretical framework. While the authors acknowledge this work, the paper does not sufficiently differentiate its contribution beyond the technical details of the approximation, raising questions about the overall novelty.

- The convergence guarantees (Theorem 5.4) are derived for the full MC approximation (Algorithm 1), but the experiments and efficient implementation rely on the surrogate approximation (Algorithm 2).

- I believe for large-scale models (e.g., SDXL, video diffusion), the cost of reward evaluation and sampling could remain significant.

---

> ### Author Rebuttal · Authors · 2026-03-31
>
> We sincerely appreciate the reviewer’s comments and suggestions. We respond below to the main weaknesses, questions, and limitations.
> > **W1: Since [1] also proposes a training-free method utilizing the same theoretical framework, the novelty of this work beyond approximation details is unclear.**
>
> **AW1:** We agree [1] uses Doob's $h$-transform as a theoretical framework for training-free diffusion adaptation. Our novelty claim is therefore not we introduce Doob's $h$-transform itself, which is a classical concept in stochastic analysis, but that we develop a novel approximation strategy for $\nabla \log h$ with a broad applicability to non-differentiable rewards.
>
> Our method estimates $h$ and $\\nabla h$ separately via MC sampling and then forms $\\nabla \\log h$ by plug-in. This does not require differentiability of the reward, so it applies to non-differentiable rewards and admits a provable convergence guarantee.
>
> By contrast, as discussed on our pages 4-5, lines 213-227, [1] uses
> $$
> \\nabla_{x_{t_l}} \\log h(x_{t_l},t_l) \\approx \\nabla_{x_{t_l}} \\log h(\\mathbb{E}[\\bar X_0 \\mid \\bar X_{t_l}=x_{t_l}],0),
> $$
> which requires differentiability of the terminal reward, equivalently of $h(\\cdot,0)$.
> > **W2 & Q1 & Q2: The convergence guarantees (Thm. 5.4) are derived for the full MC approximation, but experiments and efficient implementation rely on the surrogate approximation.**
>
> **AW2:** We agree the current theory in Thm. 5.4 is established for Alg. 1, while experiments use the surrogate implementation (Alg. 2) for computational efficiency. Alg. 2 replaces the terminal samples $x_0^{(m)}$ with the surrogate $\\hat x_0^{(m)}$, which introduces an additional approximation error not covered by Thm. 5.4 as currently stated. Our theoretical analysis extends to surrogate approximation with very minor modification as summarized in the following corollary. In particular, we additionally assume
>
> **Assumption 5.5.**
> There exists $\\varepsilon\_\{\\mathrm{surr}\}\geq 0$, such that
> $$
> \\mathbb{E}_\{X\_\{t_l\}\\sim P^h\_\{\theta,t_l\}\} \\left[\\|\\nabla  \\log \\tilde{ h}\(X\_\{t_l\},t_l\) -\\nabla \\log \\hat h(X\_\{t_l\},t_l)\\|_2^2 \\right]\\leq\\varepsilon\_\{\\mathrm{surr}\}
> $$
> holds for any $l\\in \\{1,...L\\}$, where $\\hat h$ is the MC estimator in Alg. 1 and $\\tilde h$ is the surrogate estimator in Alg. 2. This quantifies the extra error introduced by the surrogate step.
>
> **Corollary 5.6.**
> Suppose Assumptions 5.1, 5.3 and 5.5 hold and choose $\\eta_t$ as in Lemma 5.2.
> Then, with probability at least $1-\\delta$, it holds that
> $$
> \mathrm{TV}\\!\left(P_{\mathrm{data}}(\cdot \mid \mathcal E_{X_0}), \hat P_\theta^h\right)
> \lesssim
> \frac{1}{\rho}\mathrm{TV}(P_{\mathrm{data}},P_\theta)
> +
> \sqrt{
> \frac{\kappa_\sigma \log \frac{M}{\delta}\sqrt{\log(1/\delta)}}{M^{1/6}}
> +
> \varepsilon_{\mathrm{dis}}T
> +
> \varepsilon_{\mathrm{surr}}T
> },
> $$
> where $\\kappa_\sigma=\frac{T}{L}\sum_{l=1}^L \sigma_{t_l}^{-2}$.
>
> The proof follows the same template as Thm. 5.4, with a direct triangle-inequality argument introducing only one additional term, $\\varepsilon_{\\mathrm{surr}}T$, which scales linearly with $T$ and is relatively mild. Thus, while Thm. 5.4 does not directly cover Alg. 2 as stated, Alg.2 can be incorporated via Cor. 5.6.
> > **W3 & Q4: For large-scale models, the cost of reward evaluation and sampling could remain significant.**
>
> **AW3:** As discussed in Sec. 6.1, our practical DOIT algorithm (Alg. 2) incurs 0 additional NFEs compared to vanilla diffusion, since it reuses the same network forward pass for the one-step surrogate approximation (Eq.10, line 347). Therefore, its sampling cost remains comparable to vanilla generation, regardless of the scale or choice of the base model. As for reward evaluation, its cost is also not affected by scaling up the base model, as long as the same reward model or criterion is used.
> > **Q3: What is the actual overhead relative to vanilla diffusion sampling?**
>
> **AQ3:** As reported in Table 3 (line 418–420), vanilla generation takes 1.3s per sample, while DOIT takes 1.7s on average. Overall, the overhead is modest and remains comparable to vanilla generation, since DOIT introduces zero additional NFEs by reusing the same forward pass, as explained in AW3.
> > **L1: The method may be sensitive to rare high-reward regions, which can cause high approximation variance.**
>
> **AL1:**  We agree the MC approximation can have high variance when the target high-reward region is extremely rare. However, when the event is rare, it is a fundamentally difficult task for inference-time adaptation. This is well understood in offline reinforcement learning through the lens of exploration coverage ([2]). We partially mitigate this via antithetic sampling, which reduces variance empirically but does not remove fundamental difficulty.
>
> [1] Nguyen et al. "h-Edit: Effective and Flexible Diffusion-Based Editing via Doob's h-Transform."
>
> [2] Jin et al., “Is pessimism provably efficient for offline RL?”

---

> > ### Author Rebuttal · Reviewer_G14x · 2026-04-06
> >
> > Thank you for the detailed and helpful rebuttal. I appreciate the clarifications on the novelty relative to prior Doob’s $h$-transform-based methods, the extension of the theory toward the surrogate approximation, and the discussion of runtime overhead. The rebuttal strengthens the paper and addresses my main questions, and I remain positive on the paper.

---

> > > ### Author Response · Authors · 2026-04-06
> > >
> > > We sincerely thank the reviewer for the careful reading and for the positive assessment after rebuttal. We are glad that the clarifications on novelty, the surrogate approximation, and runtime overhead helped address the main concerns, and we greatly appreciate the reviewer’s recognition that these clarifications strengthen the paper.

---

### Decision · Program_Chairs · 2026-04-30

**Decision:**

Accept (regular)

**Comment:**

The reviewers agree that this submission studies an important problem in training-free adaptation of diffusion models under generic, potentially non-differentiable rewards, and they recognize several clear strengths of the work. Specifically, reviewers highlighted the use of Doob’s h-transform as a principled framework for reward-conditioned adaptation, the value of providing a convergence analysis for the prototypical Monte Carlo-based method, the practical surrogate design that substantially reduces computation, and the empirical performance on D4RL and related experiments. However, the discussion also identified concerns.

The main concern, raised most strongly by one reviewer and acknowledged in different forms by others, is that the theoretical analysis is developed for the prototypical algorithm, whereas the experiments rely on a practical surrogate implementation. Therefore, the manuscript does not currently separate clearly enough what is formally justified from what is supported primarily by empirical evidence. Additional less major concerns were raised about the extent of novelty relative to prior Doob’s h-transform-based work and about the method’s sensitivity in rare-event regimes.

I carefully read the rebuttal and follow-up discussion. The rebuttal was helpful and clarified several points, including the distinction between the principled prototype and the practical surrogate, the comparison to rejection sampling and best-of-K-style baselines, and the paper’s novelty relative to prior work. These clarifications were sufficient for several reviewers to remain positive, and one reviewer explicitly raised their score after rebuttal. Overall, reviewers appreciated the paper’s strong technical contributions, clear motivation, and thoughtful empirical evaluation, while also raising concerns about the consistency between the theoretical analysis and the empirical method.